# Process-evaluation of forest aerosol-cloud-climate feedback shows clear evidence from observations and large uncertainty in models

Sara M. Blichner [1,2] ✉, Taina Yli-Juuti [3], Tero Mielonen [4], Christopher Pöhlker [5], Eemeli Holopainen [3,4,14], Liine Heikkinen [1,2], Claudia Mohr[1,2,15,16], Paulo Artaxo [6], Samara Carbone[7], Bruno Backes Meller [6], Cléo Quaresma Dias-Júnior [8], Markku Kulmala [9], Tuukka Petäjä [9], Catherine E. Scott [10], Carl Svenhag[11], Lars Nieradzik [12], Moa Sporre [11], Daniel G. Partridge [13], Emanuele Tovazzi[13], Annele Virtanen [3], Harri Kokkola [3,4] & Ilona Riipinen [1,2]

Natural aerosol feedbacks are expected to become more important in the future, as anthropogenic aerosol emissions decrease due to air quality policy. One such feedback is initiated by the increase in biogenic volatile organic compound (BVOC) emissions with higher temperatures, leading to higher secondary organic aerosol (SOA) production and a cooling of the surface via impacts on cloud radiative properties. Motivated by the considerable spread in feedback strength in Earth System Models (ESMs), we here use two long-term observational datasets from boreal and tropical forests, together with satellite data, for a process-based evaluation of the BVOC-aerosol-cloud feedback in four ESMs. The model evaluation shows that the weakest modelled feedback estimates can likely be excluded, but highlights compensating errors making it difficult to draw conclusions of the strongest estimates. Overall, the method of evaluating along process chains shows promise in pin-pointing sources of uncertainty and constraining modelled aerosol feedbacks.

Since the industrial revolution, the total anthropogenic warming caused by increased greenhouse gas concentrations in the atmosphere has been partially masked by the cooling from anthropogenic emissions of aerosol particles and their precursors[1]. Atmospheric aerosol particles scatter and absorb incoming radiation directly (aerosol-radiation interactions, ARI) and act as cloud condensation nuclei (CCN). Increased aerosol loadings are thus expected to lead to brighter and more reflective clouds (aerosol-cloud interactions, ACI)[2,3]. In the assessments of historic climate forcing and projections of future climates, ACI have remained among the largest sources of uncertainty[4].

[1]Stockholm University, Department of Environmental Science, Stockholm SE-106 91, Sweden. [2]Stockholm University, Bolin Centre for Climate Research, Stockholm, Sweden. [3]University of Eastern Finland, Department of Technical Physics, 70211 Kuopio, Finland. [4]Finnish Meteorological Institute, Kuopio FI-70211, Finland. [5]Max Planck Institute for Chemistry, Multiphase Chemistry Dept., 55128 Mainz, Germany. [6]Universidade de Sao Paulo, Instituto de Fisica, 05508–090 Sao Paulo, Brazil. [7]Federal University of Uberlândia, Institute of Agrarian Sciences, Uberlândia, MG, Brazil. [8]Federal Institute of Pará, Department of Physics, Belém, Pará, Brazil. [9]University of Helsinki, Institute for Atmospheric and Earth System Research (INAR), Helsinki, Finland. [10]University of Leeds, School of Earth and Environment, Leeds LS2 9JT, UK. [11]Lund University, Department of Physics, 221–00 Lund, Sweden. [12]Lund University, Dept of Physical Geography and Ecosystem Science, 221–00 Lund, Sweden. [13]University of Exeter, Department of Mathematics and Statistics, Exeter, United Kingdom. [14]Present address: Institute for Chemical Engineering Sciences, Foundation for Research and Technology - Hellas (FORTH/ICE-HT), Patras, Greece. [15]Present address: Department of Environmental System Science, ETH Zurich, Zurich, Switzerland. [16]Present address: Paul Scherrer Institute, Villigen, Switzerland. ✉e-mail: sara.blichner@aces.su.se

One of the factors contributing to this is the non-linear response of ACI to perturbations in particle emissions, which is stronger in a clean than a polluted atmosphere[3,5]. The importance of natural aerosols and the feedbacks associated with them may hence increase (again) as we move into a warmer future where air pollution mitigation is expected to give a cleaner atmosphere and thus a reduced aerosol cooling[1]. To accurately capture these effects, reliable representation and evaluation of natural aerosol feedbacks in Earth System Models (ESMs) are needed.

The biogenic secondary organic aerosol (BSOA) driven feedback is one of the natural feedback mechanisms that has been proposed to compensate part of the reduction in anthropogenic aerosol emissions[6–9]. This feedback is initiated by the strong positive relationship between temperature and the emissions of BVOCs[10–12], which form SOA after being oxidised in the atmosphere. The enhanced production of SOA can then cool the surface by increasing the aerosol optical depth (ARI related) and the number concentration of CCN (ACI related)[6–8,13]. Despite considerable research effort being put into BSOA feedback modelling, the estimates of the feedback strength vary by two orders of magnitude – from highly significant ( $-0.28$ Wm$^{-2}$ K$^{-1}$ for NorESM2, offsetting almost 13% of the forcing from a doubling of CO$_2$ from pre-industrial levels[14]), to completely negligible (0.001 Wm$^{-2}$ K$^{-1}$ for UKESM1 in[15], Table 9) see also[8,9,13,16].

Boreal and tropical forests currently constitute about 27 and 45 % of global forested area, respectively[17]. These forest ecosystems are among the greatest sources of BVOCs emitted to the atmosphere[18] and the total global SOA burden[18,19], and are therefore important drivers of potential BVOC feedbacks. Tropical forests are characterised by high diversity in tree species[20], while boreal forests have fewer species including a larger fraction of coniferous trees[17,21]. This leads to a different spectrum of BVOCs emitted by these two ecosystems: the tropical BVOC emissions are dominated by isoprene (IP), while monoterpenes (MT) typically dominate the VOCs emitted from boreal forests[22–25]. The differences between the tropical versus the boreal forest range from drivers of BVOC emissions[10,26], the molecular spectra of the emitted species, oxidation chemistry e.g.[27], the hydrological cycle, to cloud regimes, and it is therefore vital to analyse both environments to understand the full impact of any feedback.

The models of SOA formation from BVOCs implemented in most state-of-the-art ESMs are relatively similar: First, BVOC emissions are calculated based on land use, vegetation and environmental conditions see e.g.[10]. Emissions are often expected, at least partially, to follow an exponential relationship with temperature see e.g.[10]. The BVOCs are then oxidised in the atmosphere by the common oxidants such as OH, ozone and NO$_3$, and some percentage (yield) of the oxidation products are assumed to form SOA. Constant yields are often used for simplicity, hence ignoring e.g. variations in nitrogen oxides' concentrations, relative humidity or aerosol acidity as well as sub-grid co-variability of oxidants and BVOCs[19]. The oxidation products of the BVOCs are lumped into some number of tracers (often two) with different representative molecular properties (e.g. volatility) that affect their behaviour such as participation in new particle formation (NPF). Models have very different degrees of sophistication in the representation of SOA formation and its aerosol particle size distribution dynamics, resulting in high inter-model variability[28,29]. ESMs need to strike a balance between process detail and computational burden. This raises the question of what degree of simplification is acceptable and what improvements in ESMs should be in focus to ensure accurate enough predictions of the BVOC-aerosol-cloud-climate feedback.

The current common practices for estimating the BVOC-aerosol-cloud-climate feedback strength in models see e.g.[14,15] are based on highly unrealistic perturbations, which cannot be evaluated against observations. However, the emergence of long-term in-situ observational data sets[30–36] gives rise to a unique opportunity to use natural

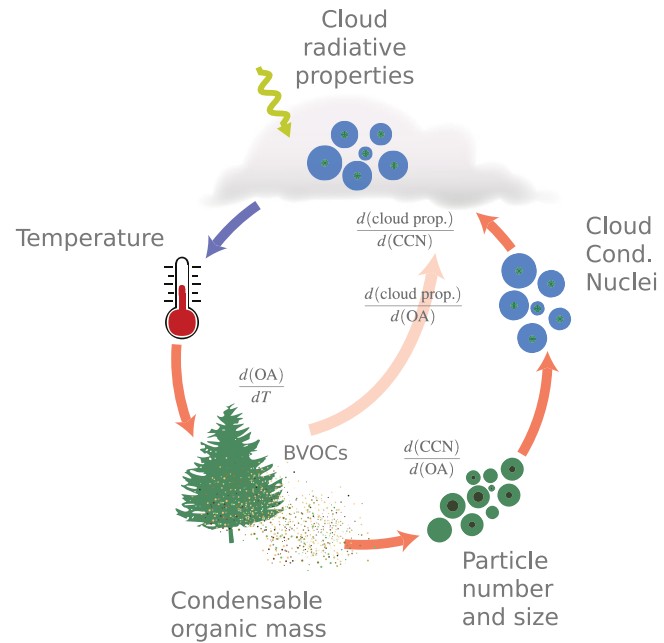

**Fig. 1 | Illustration of the key parameters and processes associated with the Biogenic Volatile Organic Compound (BVOC)-aerosol-cloud-climate-feedback.** Red arrows indicate positive (enhancing) and blue negative (dampening) responses.

variability in environmental parameters and aerosols as a proxy for perturbed states of the climate. While satellite data has been used to evaluate e.g. cloud processes in ESMs[37–39], in-situ data has mainly been used to evaluate state variables in aerosols (mass and number concentrations, size distribution etc.), but not the relationships or processes connecting these. Combined with satellite data, these in-situ data sets make it possible to follow regional processes affecting aerosol composition and size distribution all the way to aerosol-cloud interactions, and to compare the modelled and observed relationships.

Here we use two emerging long-term in-situ data sets representing the boreal and tropical forest environments together with satellite data to evaluate ESMs with respect to the modelled relationships between the variables in the feedback loop depicted in Fig. 1. The Station for Measuring Ecosystem-Atmosphere Relationships (SMEAR-II)[40] in Southern Finland is used for the boreal zone and the Amazon Tall Tower Observatory (ATTO) measurement station is used to represent a tropical rainforest environment[41]. We evaluate the models by examining the components of the BVOC-aerosol-climate feedback chain from the temperature dependence of emissions through to subsequent impacts on organic aerosol mass, particle number size distribution and cloud properties (following[8]) using natural variability in environmental conditions as a proxy for a perturbed climate state: i.e. we evaluate the relationships between the variables in the feedback loop under natural variability of weather. We seek to gain insight into the key processes contributing to the BVOC-aerosol-cloud-climate feedback strength and behaviour in these two globally important forest ecosystems, and their representation in ESMs. The total feedback is by definition the change in radiative forcing ($F$) with temperature ($T$) and can be decomposed as follows:

$$\frac{dF}{dT} = \frac{d(OA)}{dT} \cdot \frac{d(CCN)}{d(OA)} \cdot \frac{d(\text{cloud prop.})}{d(CCN)} \cdot \frac{dF}{d(\text{cloud prop.})} \quad (1)$$

where CCN is cloud condensation nuclei concentration, OA is the organic aerosol mass, and "cloud prop." refers to cloud properties. In this study, we target the terms in the feedback up until changes in

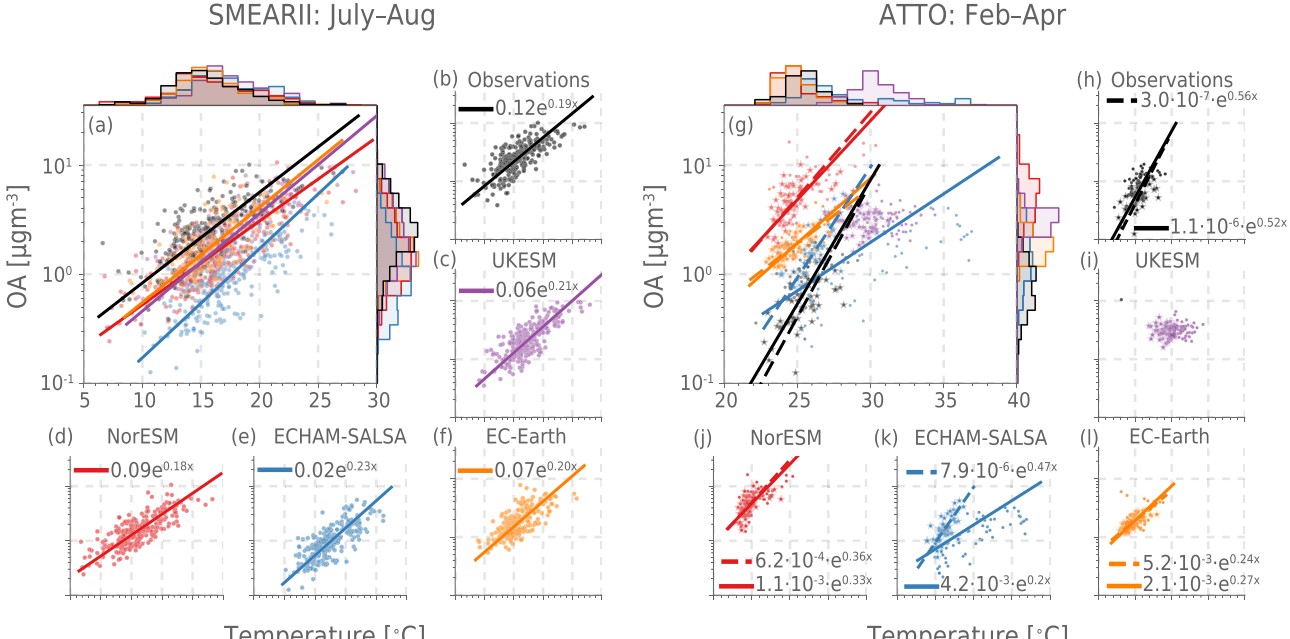

**Fig. 2 | Relationship between temperature and organic aerosol (OA) mass concentration in Earth System Models (ESMs) (blue, red, orange and purple for ECHAM-SALSA, NorESM, EC-Earth and UKESM, respectively) and observations (black) at The Station for Measuring Ecosystem-Atmosphere Relationships (SMEAR-II, a–f) and Amazon Tall Tower Observatory (ATTO, g–l) during periods where biogenic secondary OA is known to dominate the OA budget (July–August in SMEAR-II and February–April at ATTO).** The main plot for each station shows all the daily median values of temperature and OA mass and the orthogonal distance regression for ln(OA) = aT + b for the observations and nudged model predictions for the same periods (see Methods for details). Histograms of the observed and predicted values are shown on the top (for temperature) and right (for OA) side of the main plots (**a** and **g**). The smaller plots display the same information as the large plot but for each data source separately. The ESM grid box covering the station (nearest-neighbour) is chosen for evaluation. Regressions are not shown for UKESM at ATTO because the correlation is too weak ($r^2 = 0.06$, see Table S2). For ATTO, star symbols and dashed lines indicate values when 2015/2016 are excluded from the data. Residuals for different regressions are shown in S19 and S20. See Tables S1–S3 for uncertainties and $r^2$ values for the regressions. Source data are provided as a Source Data file.

cloud properties, i.e.

$$\frac{d(\text{cloud prop.})}{dT} = \frac{d(\text{OA})}{dT} \cdot \frac{d(\text{CCN})}{d(\text{OA})} \cdot \frac{d(\text{cloud prop.})}{d(\text{CCN})}$$
$$= \frac{d(\text{OA})}{dT} \cdot \frac{d(\text{cloud prop.})}{d(\text{OA})} \qquad (2)$$

The relationship between changes in cloud properties and forcing $\left(\frac{dF}{d(\text{cloud prop.})}\right)$ is currently an active area of research in its own right see e.g.[3,42], which we, therefore, leave outside of the scope of this study. Note that using present-day conditions to evaluate the feedback limits our analysis to the "pure" temperature feedback and excludes the potential effect of $CO_2$ fertilisation on BVOC emissions, which has been shown to be a large contributor in some ESMs see e.g.[43]. We use concentrations of number of particles larger than 50 nm, 100 nm and 200 nm ($N_{50}$, $N_{100}$, $N_{200}$) to investigate the link between OA and CCN, "cloud prop." is investigated by changes in cloud droplet effective radius and cloud optical thickness, and we evaluate the terms $\frac{d(\text{CCN})}{d(\text{OA})}$ and $\frac{d(\text{cloud prop.})}{d(\text{CCN})}$ both separately and combined, i.e. $\frac{d(\text{cloud prop.})}{d\text{OA}}$, to assess the combined effect. In our process-based evaluation, we combine the insights from relevant observational data sets with the unrivalled ability of ESMs to produce projections on a global scale.

## Results
### Relationship between temperature and organic aerosol mass concentration

Observational data from both SMEAR-II and ATTO display a clear positive relationship between temperature and organic aerosol (OA) mass concentrations (Fig. 2), described by an exponential function of the form SOA = $\alpha \exp(\beta T)$ (residuals in Figs. S19 and S20). This is in line with the expected exponential relationship between temperature and emissions of BVOCs[10,11] assuming the availability of BVOCs is a major factor controlling SOA formation at the two sites compared to other factors[8]. In the equation, $\alpha$ thus incorporates factors related to the baseline emission strength of BVOCs, SOA yield, and the loss rates of SOA, while the $\beta$ term is related to the temperature dependency (including possible temperature dependencies of yields, oxidation, loss rates etc.). At SMEAR-II, the models all underestimate OA mass concentration (see also section S9.1). However, $\beta$ (the slope of the line in the linear-log space in Fig. 2) is surprisingly well represented in all the models, in spite of different emission schemes for BVOCs (see Table 1). This means that at SMEAR-II, the models would agree with observations very well if SOA yields were scaled up, all else being equal. For ATTO, on the other hand, the models display very different behaviour from each other and the observations, UKESM even showing no real relationship ($r^2 = 0.06$ for $T$ versus ln(OA)) between OA with temperature. Note that both ECHAM-SALSA and UKESM have anomalously high temperatures for the years 2015 and 2016 (see discussion in section S4.2 and in particular Figs. S11 and S14). If these anomalous years are excluded, then ECHAM-SALSA is closer to the observations because these years have highly abnormal emissions (the $\beta$-term in the observations is 0.56 and ECHAM-SALSA has 0.20 with 2015/2016 and 0.46 without), while for UKESM the correlation stays similarly low as with these years (see Table S3). The $\beta$-term in NorESM and EC-Earth both have smaller values than the observed (0.33 and 0.25 versus the observed 0.56), but this comes with a large overestimation of the OA mass concentration, especially for NorESM. Contrary to SMEAR-II, the OA concentrations at ATTO in the models are consistently too high,

**Table 1 | Summary of model components**

| Model | NorESM[61] | ECHAM -SALSA[65] | EC-Earth[62,63] | UKESM[64] |
|---|---|---|---|---|
| BVOC emissions | MEGAN2.1[10] | MEGAN2.1[10,68] | LPJ-GUESS 4.1[47,69] | iBVOC: ref. [66,67] IP from[70] MT from[48] |
| Aerosol scheme | OsloAero6[71] (hybrid/plume scheme) | SALSA[65] (sectional) | M7[72] (modal) | UKCA-GLOMAP[73] (modal) |
| Vegetation/land surface model | CLM-BGC[74] | JSBACH v3[77–79] | H-TESSEL[80] | JULES[75,76] |
| Activation scheme | Abdul-Razzak & Ghan(ARG)[82] | ARG[81] sectional | ARG[82] with updraft pdf | ARG[82] with updraft pdf |
| Nucleation | BL: ref. [83] Everywhere: ref. [84] | Strat: ref. [84] Trop: ref. [86] | BL: ref. [85] Everywhere: ref. [84] | Everywhere: ref. [84] |
| SOA treatment | 2 products Cond. as non-volatile | 3 bin VBS Cond. as semi-volatile | 2 products Cond. as non-volatile | 1 product Cond. as non-volatile |
| Warm clouds | Double-moment bulk scheme MG2[87] | Double-moment bulk scheme[88] | Single-moment bulk scheme[63] | Single-moment bulk scheme[73] |

"Cond. as non-volatile" means the organic products are treated essentially as non-volatile during condensation. The nucleation rates are reported together with which part of the atmosphere they are applied to: only in the boundary layer (BL), in the troposphere (Trop.), in the stratosphere (Strat.) or everywhere.

with NorESM being the most extreme (Fig. 2 and S49). Note also that the temperature range at ATTO is much smaller than at SMEAR-II. This is a natural consequence of a tropical versus a boreal environment and introduces some more noise into the signal in ATTO than in SMEAR-II.

These results confirm step (1) in our simplified chain of processes (Fig. 1), i.e. that temperature indeed is a key factor regulating atmospheric OA for both boreal and tropical forest environments in seasons where we expect biogenic SOA to dominate the OA budget. The studied ESMs reproduce this relationship reasonably well in the boreal zone, but in the tropical environment there is a large disparity between the models, with no model seeming to reasonably capture OA formation and its relationship to temperature.

In view of the results above, it might seem tempting to tune the IP and MT SOA yields to improve the modelled OA concentrations, given the dominance of MT in the boreal zone and IP in the tropics. However, further investigations with NorESM reveal that with the current model setup, this tuning problem might not have a solution (the best fit to the observations has negative IP yields) and improving the model may require more sophisticated yield calculations and the related oxidation chemistry (see section S8.1.1).

**Relationship between organic aerosol mass and number/size of particles**

The relationship between OA mass concentration and particle number concentration in different size ranges indicates which parts of the aerosol particle size distribution are perturbed when the production of SOA is changed, and thus how strong the change in CCN and cloud properties might be due to increasing BVOC emissions in the chain of processes depicted in Fig. 1. If, for example, SOA mainly condenses onto coarse mode particles (larger than $2.5\,\mu m$ in diameter), then the climate effects may be weak even with strong changes in emissions, because these particles are already large enough to act as CCN. On the other hand, if there is a significant addition of new particles in the CCN range (particles larger than the activation diameter which is typically 50 nm to 200 nm, depending on particle composition and ambient maximum supersaturation) then the climate impact may be very large even for relatively modest emission enhancements.

The observational results shown in Fig. 3 (see Tables S1–S2 for regression details) confirm a clear positive relationship between OA and $N_{50}$, $N_{100}$ and $N_{200}$ for both SMEAR-II and ATTO, which is mostly represented in the models as well. However, the comparison reveals a great spread in how the models represent the sensitivity of particles large enough to act as CCN to perturbations in OA mass. Since SOA dominates the OA budget at these locations and times, this describes the sensitivity of climate-relevant particle number concentrations to

SOA formation. For example at ATTO, EC-Earth, ECHAM-SALSA and UKESM underestimate the slope between OA and $N_{100}$, while NorESM overestimates it (see Tables S1–S2). NorESM stands out as strongly overestimating the impact of OA on $N_{100}$ and $N_{50}$, while simultaneously underestimating the impact on $N_{200}$. This is likely related to a fairly high concentration of Aitken mode particles in NorESM see e.g.[29] and, consequently, a large perturbation resulting from condensing SOA growing these particles. UKESM and EC-Earth perform very similarly and are the models closest to capturing the impact on $N_{200}$, while both ECHAM-SALSA and NorESM underestimate this. Overall, EC-Earth, ECHAM-SALSA and UKESM consistently underestimate the impact of OA on $N_{100}$ and $N_{50}$, while NorESM overestimates it .

At SMEAR-II observations reveal a distinct buffering in the impact on $N_{50}$ and $N_{100}$ for high OA concentrations and the relationship is well captured by a logarithmic function, $N_x = a + b \ln(c + OA)$. This buffering is not seen in the same way at ATTO. A buffering could be expected for many different reasons: for one, with increased OA and thus number concentration, the organic vapours will condense onto more particles, thus limiting the growth of each particle per increase in OA. Secondly, high loadings of OA could inhibit new particle formation (NPF) which will be suppressed both by an increased condensation sink (reducing pre-cursor concentrations) and coagulation sink (reducing survival of particles to larger sizes) e.g.[44]. SMEAR-II is known to have frequent NPF, even in summer[45], ATTO is known to have very little[22], which could explain the discrepancy between the two stations. On the other hand, it could also be due to the aerosol concentrations (both number and mass) simply being too low at ATTO compared to SMEAR-II, so that the buffering would only be seen at higher concentrations. The buffering seems to be captured in all models except NorESM, but it is too strong in EC-Earth, ECHAM-SALSA and UKESM. None of the models have buffering at ATTO. The observed relationships between OA and number concentration for the two environments (SMEAR-II and ATTO) are almost identical if both are fitted with a linear regression (see Figs. S17 and S18). The models, however, predict a larger difference between the two sites, with the exception of UKESM. The consistency of the observed slope may of course be incidental, or it might be a symptom of some process constraining the size distribution dynamics in reality which is currently not well represented in models.

Overall, these results reveal a large uncertainty in the modelled processes of SOA formation and highlight the importance of adequately capturing aerosol size distribution dynamics.

**Relationship between OA and cloud properties**

We here analyse the modelled and observed impact on cloud properties, as represented by cloud optical thickness and cloud droplet

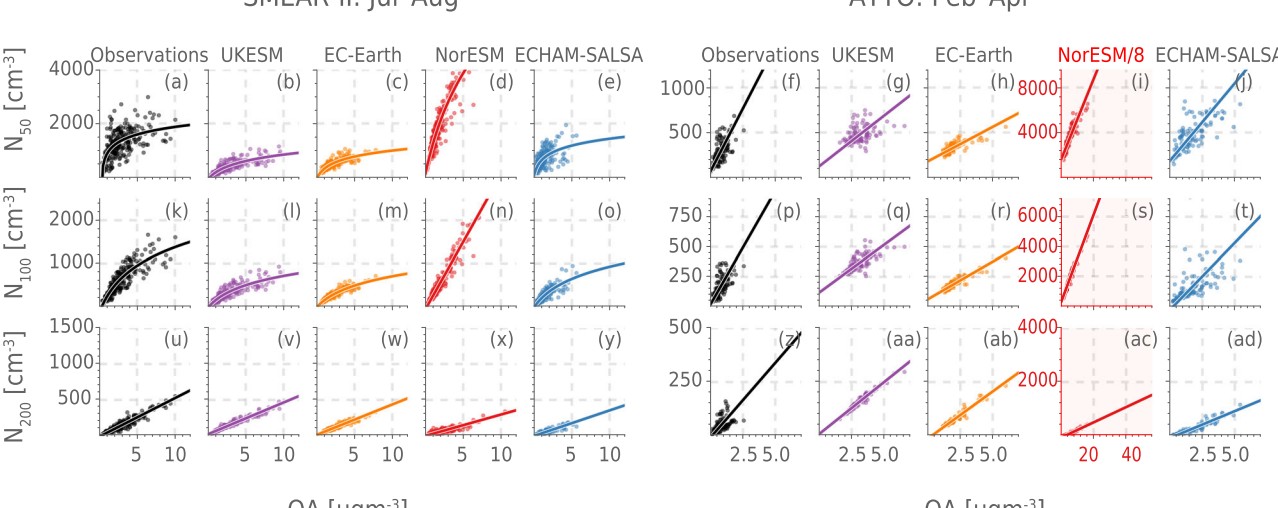

**Fig. 3 | The relationships between daily median organic aerosol (OA) mass concentration and the number concentration of particles larger than 50 nm ($N_{50}$), 100 nm ($N_{100}$) and 200 nm ($N_{200}$).** The Station for Measuring Ecosystem-Atmosphere Relationships (SMEAR-II) in July–August is shown in the left panel in **a–e** ($N_{50}$), **k–o** ($N_{100}$) and **u–y**($N_{200}$) and the Amazon Tall Tower Observatory (ATTO) in February–April is shown in the right panels in **f–j** ($N_{50}$), **p–t** ($N_{100}$) and **z–ad**($N_{200}$). For ATTO, the size distribution measurements only go up to 500 nm, so we use the same intervals for the models. The lines show the least-square regression to a logarithmic function ($a + b\ln(c + x)$) for $N_{50}$ and $N_{100}$ at SMEAR-II and the orthogonal distance regression to a linear function for all the others ($ax + b$) (see Methods for details). The regressions and their properties are summed up in Tables S1–S2 and the residuals for the regressions are shown in Figs. S21, S23, S25–S28. The equivalent figure, but with all regression lines linear is shown in Fig. S17 with residuals in Figs. S22 and S24. NB: For ATTO, the axis limits for NorESM are eight times those of the other plots (indicated by the red axis and background colour of the plot). Also note that for $N_{100}$ at SMEAR-II, NorESM is shown with a linear fit because the regression did not converge for the logarithmic function. See Tables S1–S2 for uncertainties and $r^2$ values for the regressions. Source data are provided as a Source Data file.

effective radius. By binning by cloud water path (CWP), we constrain the impact of different cloud regimes/types on our analysis and also effectively constrain it to mainly the cloud albedo effect (or the first indirect effect), leaving other aerosol-cloud interactions outside the scope of this study[3]. We focus on the change in cloud properties as a result of changes in OA (Fig. 4), not CCN. This is because the activation diameter may vary both within and between models and reality, and to evaluate the feedback strength, it is easier to follow the signal as outlined in Eq. (2) by considering $\frac{d(\text{cloud props.})}{d(\text{OA})}$ rather than $\frac{d(\text{cloud props.})}{d(\text{CCN})}$. See the supplementary, section S7.2, for the same figure as Fig. 4 but with high versus low $N_{100}$ and $N_{50}$.

The response of cloud properties to OA, as derived from satellite data from areas around both SMEAR-II and ATTO, aligns well with the simplified picture outlined in Fig. 1: the median cloud optical thickness (COT) is generally higher and cloud droplet effective radius ($r_{\text{eff}}$) lower on high OA days as compared with low OA days (Fig. 4), within the same bin of CWP. This is true for the majority of the CWP bins, although there are bins where the difference is not significant (e.g. the lowest bins of CWP at SMEAR-II and the lowest and highest in ATTO). This is, to the best of our knowledge, the first time such an analysis has been done for the tropical environment with results indicating an environment where the clouds are sensitive to changes in BVOC emissions.

The ESMs, on the other hand, do not provide a uniform picture of the response of the cloud properties to changes in OA. For SMEAR-II (Fig. 4a, c), none of the analysed models consistently replicate the observed increase in COT and decrease in $r_{\text{eff}}$ on days with high OA versus low OA. While NorESM produces the right sign for the difference in COT and $r_{\text{eff}}$, the magnitude of the response is clearly too high (more than double that of the observations for CWP below 250 gm$^{-2}$) compared to the observations, especially considering that the total OA concentrations (and thus also the change between high and low) are lower in the model than in the observations. This is likely due to the overestimation of the slope between OA and $N_{100}$ and $N_{50}$ meaning that the increase in CCN is too strong for high OA concentrations. The

same analysis for $N_{100}$ instead of OA (see Fig. S30a) also shows an equally strong overestimation, which could indicate a too strong aerosol sensitivity in general in the model. ECHAM-SALSA often is close to the observations in the median, but the uncertainties are high and the response is significantly different from zero for only very few CWP bins. The model also shows an increase in $r_{\text{eff}}$ (as opposed to the expected decrease) for the smallest CWP bins at SMEAR-II, though not significantly different from zero. These results for ECHAM-SALSA stand in contrast to the same analysis for high versus low $N_{100}$ and $N_{50}$ (Figs. S30 and S31), which show a stronger and more often significant response. It is therefore likely that the weak relationship between OA and $N_{50}$ seen in the previous section (see Fig. 3) likely plays a significant role in reducing the impact of OA on cloud properties. UKESM and EC-Earth show a significant response only for the lowest CWP bins in the boreal environment, where the observations show a very weak change in cloud properties. The low response in UKESM in the boreal zone is not due to a weak aerosol sensitivity though (see Fig. S30a), but is likely due to an erroneously low hygroscopicity of OA in UKESM (confirmed through code inspection) which counteracts the effect of size during activation. Note that the distribution of CWP in UKESM is quite heavily skewed towards lower values, meaning that the smallest bins showing a significant response in Fig. 4 for SMEAR-II actually constitute most of the data. EC-Earth has a similar lack of response in the higher CWP bins for $N_{100}$ and $N_{50}$ as for OA. This limits the impact of the feedback and is not in accordance with the observations, especially for the middle values of CWP. For ATTO, all the models underestimate the change in $r_{\text{eff}}$ with elevated OA concentrations, possibly with the exception of ECHAM-SALSA. On the other hand, for COT, NorESM is very similar to the observations, though with some non-significant overestimation for the middle CWP bins. ECHAM-SALSA overestimates the response for all CWP bins and is more than a factor of two too high for all bins above 170 gm$^{-2}$. Note that both models show inconsistent responses in $r_{\text{eff}}$ compared to COT, while the observations mostly show the response to be mirrored between the two. The low response of the modelled $r_{\text{eff}}$ for

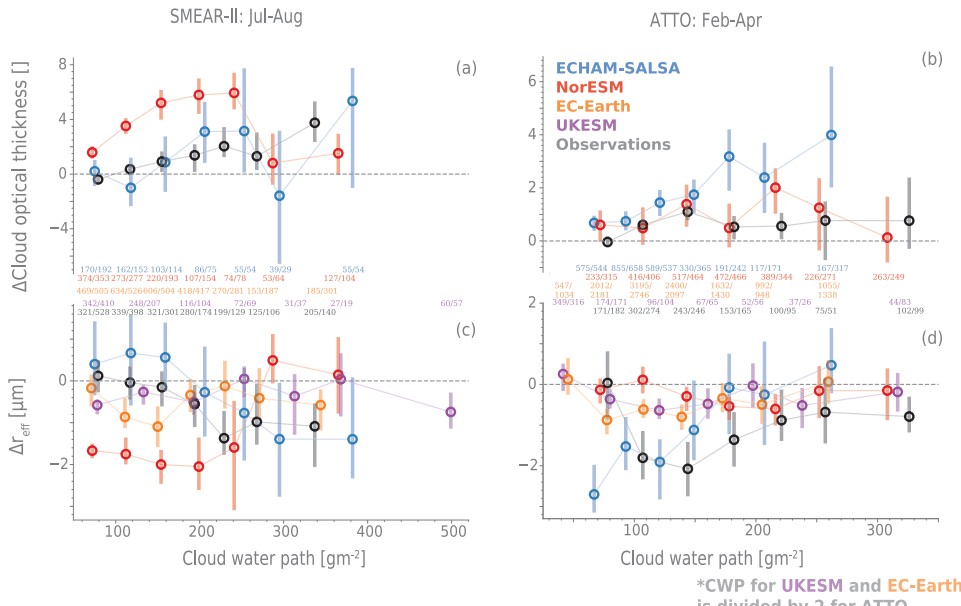

**Fig. 4 | The change in cloud properties between days with high and low organic aerosol (OA) concentrations.** The difference in median cloud optical thickness is shown in **a** (The Station for Measuring Ecosystem-Atmosphere Relationships, SMEAR-II) and **b** (Amazon Tall Tower Observatory, ATTO) and the difference in cloud droplet effective radius ($r_{eff}$) is shown in **c** (SMEAR-II) and **d** (ATTO). High OA is defined as above 67th percentile days and low OA as below 34th percentile days. The uncertainties marked by the bars and are the 5th to 95th percentile of the median calculated by bootstrapping and the percentile method (see Methods section). The numbers in the middle signify the number of data points in low/high OA in each cloud water path (CWP) bin. Note that for ATTO, UKESM and EC-Earth CWP values have been divided by two to fit on the same x-axis as the other data sources. The cloud optical thickness (COT) output was not available for EC-Earth and UKESM and these are therefore only shown in the lowermost panels. Source data are provided as a Source Data file.

all the models in ATTO might be related to droplet sizes being smaller in the models (see Fig. S34b), making the clouds less susceptible to perturbation (updraft- rather than CCN-limited see e.g.[32]). This is consistent with the response of the cloud properties to high versus low $N_{50}$ and $N_{100}$, which is similarly low as for OA, and also the fact that the models overestimate aerosol concentrations at ATTO. In NorESM in particular, the extreme overestimation of both OA and the number of particles in the region (Figs. 2 and 3), likely results in the cloud regime to be updraft- rather than CCN-limited. Note that this is the opposite picture to SMEAR-II where most models were too sensitive to changes in $N_{50}$ and $N_{100}$.

The changes in cloud properties are the most challenging both to compare and interpret due to e.g. the diversity in representation in the models, and different resolutions and uncertainties in the satellite products. This is illustrated by the relatively large differences between the models and observations in absolute distributions of CWP, $r_{eff}$ and COT shown in Figs. S32 and S34. Furthermore, the models do not have exactly the same diagnostics and vary greatly in their representation of cloudiness (see Methods section). In spite of these challenges, we believe that the results above clearly suggest at least the following: (1) the observational data from SMEAR-II and ATTO confirm the final step in the proposed feedback chain (Fig 1) in terms of the key dependencies driving the BVOC-aerosol-cloud-climate feedback for liquid-phase clouds; (2) NorESM overestimates the strength of the feedback in the boreal zone; (3) most of the models underestimate the impact on $r_{eff}$ over the Amazon, thus reducing the feedback strength in these models in this environment; (4) overall, the ESMs do not provide a consistent picture of the COT and $r_{eff}$ response to OA — underlining the need for improved constraints on this crucial processes in the feedback loop.

## Discussion

We show that the proposed feedback in Fig. 1 is detectable in observations and likely plays an important role in tropical (presented for the

first time here) and boreal forests (already presented in ref. 8). Furthermore, our methodology for evaluating feedbacks in ESMs, using the interplay between key variables in the present-day climate to evaluate how realistic a given feedback is represented in ESMs, is shown to be efficient in identifying model weaknesses and also compensating errors along the process chain.

As mentioned, our evaluation only targets the temperature feedback and not the direct effects increased $CO_2$ concentrations can have on BVOC emissions – both through the so-called $CO_2$ fertilisation effect[46] and through $CO_2$ inhibiting isoprene emissions directly see e.g.[47]. This $CO_2$ "branch" is not strictly a feedback since there is no dependency on temperature, but is still highly relevant for future emissions and is included in the feedback estimates in[15]. Gomez et al.[43] show that for the latest generation of ESMs (Coupled Model Intercomparison Project 6, CMIP6), the $CO_2$ branch is highly variable, and again UKESM and NorESM stand out as extremes: in UKESM the $CO_2$ branch has a negative effect on BVOC emissions due to $CO_2$ inhibition while in NorESM the $CO_2$ fertilisation dominates and the signal of the $CO_2$ branch is slightly higher than the temperature branch. In spite of this, most of the difference between UKESM and NorESM originates in the conversion of the emission change to actual forcing, as can be seen in Table 9 in ref. 15. The total change in emissions per change in temperature (which includes $4 \times CO_2$) is a factor of approximately 7 larger for NorESM than UKESM, while the change in effective radiative forcing (ERF) per change in emissions is a factor of 30 larger for NorESM than for UKESM. Our analysis is able to pinpoint specific issues in both models in exactly the steps that convert changing emissions to forcing. Based on the four ESMs in this study, we demonstrate especially the following four issues:

1. While the models generally get the temperature dependence of OA mass concentration right for boreal forests (SMEAR-II), for tropical forests (ATTO), both the dependency on temperature

and the overall OA mass is highly diverse and in particular UKESM has no response to temperature at ATTO.

2. The influence of OA on particle number and size in the models is very diverse, which suggests that size distribution dynamics are highly important for the feedback strength.

3. These relationships between OA mass concentration and particle number concentration affect the strength of the OA impact on clouds.

4. The hygroscopicity of OA in the models may affect the total feedback. This is seen for UKESM in the boreal zone, where we find a relatively weak cloud response to OA, in spite of the model being quite sensitive to $N_{100}$ (similar to EC-Earth and ECHAM-SALSA).

The models perform overall much more coherently with both each other and the observations in SMEAR-II compared to ATTO. This is perhaps not surprising: a long history of both forest and aerosol research performed at SMEAR-II means that the observations from this station have been used actively during the development of these models and components therein. It does, however, speak to the importance of both establishing long-term measurements in under-sampled parts of the atmosphere – especially in the southern hemisphere – and placing more weight on already existing measurements, like those from ATTO, in model development. It is, for example, possible that the fixed SOA yield approach used in these models works well enough in a boreal forest, but fails in a high isoprene environment like the tropics see e.g.[27]. If so, this is a significant structural error in the models.

In this study, two of the models have interactive oxidant chemistry, UKESM and EC-Earth. This is in contrast to NorESM and ECHAM-SALSA, where oxidant concentrations are read from file and cannot be affected (e.g. depleted) by changes in the BVOC emissions. It is interesting that these models both have too low a slope or a too weak relationship between temperature and OA mass (Fig. 2). This is especially true for UKESM, where the temperature dependency of monoterpene emissions is very similar to NorESM and ECHAM-SALSA (they all use MEGAN or pre-runners[10,48]), meaning that the dependency of OA mass on temperature is lost during the oxidation process. A recent study by[49] using UKESM emphasises that oxidant chemistry can play a major role in the total feedback, also for the cloud-aerosol interactions. While the oxidant chemistry response is clearly missing in models like NorESM and ECHAM-SALSA that have fixed oxidant concentrations, the fact that the models in this study with interactive chemistry (EC-Earth and UKESM) agree poorly with the observations for the temperature to OA relationship might encourage further improvements of the oxidant chemistry in the tropical forest regions in the models.

Reliable climate projections, with natural feedbacks being included, require a sufficiently accurate representation of fundamental processes and their response to changes in emissions or other forcers. Our results show that evaluating mean absolute variables produced by a model, although generally a common practice within climate science, is not alone a sufficient measure for how well a model performs in this task. As an example, NorESM performs reasonably in representing $N_{100}$ concentrations at SMEAR-II (see e.g. Fig. S41), but overestimates its response to changes in OA (see Fig. 3). The predictive power of a given model depends on the robustness of the underlying theoretical treatment in reproducing the inter-dependencies of the key variables. As demonstrated here for the BVOC-aerosol-cloud-climate feedback, the combination of emerging long-term in-situ measurements, satellite data, and process understanding bear great potential in finding new ways to evaluate and constrain ESMs, and reduce uncertainties in their projections.

Of the models considered here, NorESM has previously been shown to have a strong feedback (around $-0.3\,\mathrm{Wm^{-2}\,K^{-1}}$), while UKESM has been shown to have a negligible feedback $0.001\,\mathrm{Wm^{-2}\,K^{-1}}$[14,15]. Estimates of the feedback strength in impacting cloud

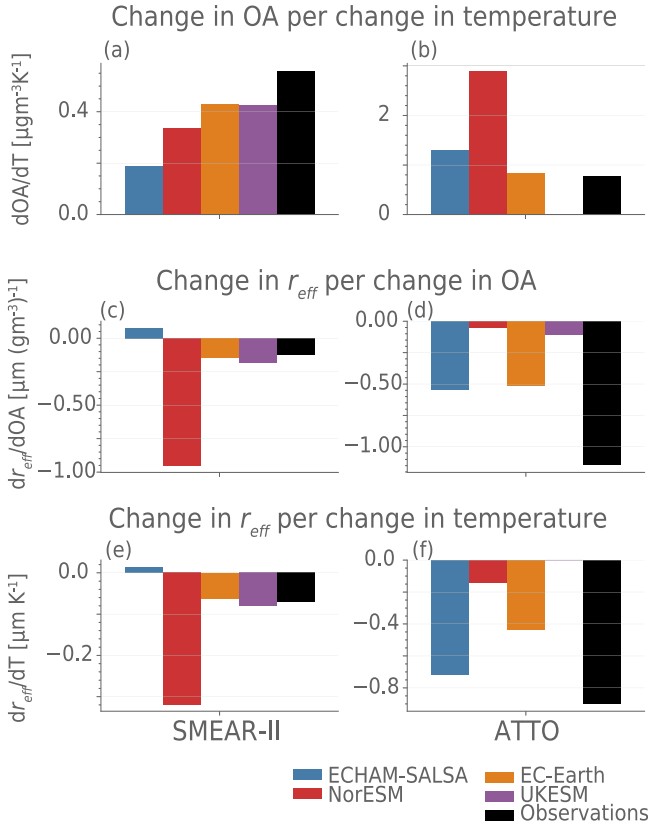

**Fig. 5 | Estimated strength of the terms in the feedback loop at the two stations.** The change in organic aerosol (OA) per temperature is shown in panel **a** (The Station for Measuring Ecosystem-Atmosphere Relationships, SMEAR-II) and **b** (Amazon Tall Tower Observatory, ATTO), the change in cloud droplet effective radius ($r_{\mathrm{eff}}$) per change in OA is shown in (**c**) (SMEAR-II) and **d** (ATTO), and finally the product of these two terms, $\frac{dr_{\mathrm{eff}}}{dT} = \frac{dOA}{dT} \cdot \frac{dr_{\mathrm{eff}}}{dOA}$ is shown in **e** (SMEAR-II) and **f** (ATTO). The change in OA per change in temperature is estimated in accordance with the exponential fit for each model evaluated based on a 3 °C temperature perturbation from the current median (in accordance with the best estimate for climate sensitivity[4]). The change in $r_{\mathrm{eff}}$ from change in OA is estimated by doing a weighted average over the bins in Fig. 4. Finally the change in $r_{\mathrm{eff}}$ by change in temperature is the product of the two above ($\frac{r_{\mathrm{eff}}}{T} = \frac{OA}{T}\frac{r_{\mathrm{eff}}}{OA}$). See Fig. S6 for the same figure with cloud optical thickness (COT). Source data are provided as a Source Data file.

properties, here represented by $r_{\mathrm{eff}}$, in this study (Fig. 5) suggest NorESM overestimates the feedback strength in the boreal zone, while displaying a much too weak feedback in the tropical region in spite of a very strong OA perturbation. On the other hand, UKESM does not simulate a feedback in the tropical zone, due to the lack of relationship between temperature and OA. ECHAM-SALSA shows a positive full feedback in SMEAR-II for $r_{\mathrm{eff}}$, but Fig. S6 shows that for COT, the full feedback strength in ECHAM-SALSA is very close to the observed estimate, thus revealing inconsistency in the modelled cloud response. Overall, our study seems to rule out the lowest model estimates of the feedback (UKESM). Although the strongest model estimates (NorESM) is revealed to overestimate the feedback in the boreal zone, the compensating error in the tropics makes it hard to completely rule it out, especially considering the important role the tropics play in the global radiation budget. The failure of the models to correctly represent the processes in the feedback chain is concerning, not only for their ability to capture this feedback, but their ability to capture aerosol–climate interactions in general. Using process–based evaluation to improve the models may aid in reducing uncertainty and spread in future climate projections. As global temperatures rise, the ability of ESMs to capture

the strength and sign of feedbacks between the biosphere and climate will make an increasingly important contribution to their ability to accurately simulate future climate.

## Methods

At each station, we focus our analysis on periods where biogenic SOA is expected to dominate the OA budget and we further limit the period in time to avoid seasonality significantly influencing the result. For SMEAR-II, we focus on July and August, as already investigated in ref. 8. For ATTO we focus on February – April, when (1) biogenic SOA is expected to dominate the OA (wet season)[22] and (2) the influence of seasonal changes in cloud properties and OA are weak (see Fig. S29). Note that there is still an increase in $r_{eff}$ over Feb–Apr. However, we find that our results do not change in character depending on which months are chosen, as can be seen in Figs. S1–S3 which show results for ATTO from similar analysis as presented in Figs. 2–4 but for different choice of months.

### Observational station data

For organic aerosol mass, we use Aerosol Chemical Speciation Monitor (ACSM; ref. 50) measurements performed within the boreal forest canopy at SMEAR-II station in Hyytiälä, Finland (4 m above ground[30]), and over the tropical forest canopy (60 m above ground and about 20 m above canopy top) at ATTO station. The OA data from SMEAR-II and ATTO covered years 2012–2018 and 2014–2018, respectively (see Fig. S36). The particle number size distribution was measured within the boreal forest canopy at SMEAR-II with a Differential Mobility Particle Sizer (DMPS)[51]. At ATTO, the particle number size distribution was measured with a Scanning Mobility Particle Sizer (SMPS), and sampling was conducted 60 m above ground level[34,52]. Number concentrations $N_{50}$, $N_{100}$ and $N_{200}$ were calculated by first interpolating the size distribution linearly to a finer resolution in diameter and then integrating over the size distribution. For SMEAR-II, the size distribution measurements go up until 1000 nm and $N_x$ thus refers to the concentration of particles between $x$ nm and 1000 nm. For ATTO, the size distribution measurements only go up to 500 nm, so $N_x$ here refers to the concentration of particles between $x$ nm and 500 nm. The same intervals are used for the models. Note that to avoid impact of diurnal cycle, we use only data where we have 15 or more observations per day (hourly resolution).

We use temperature and wind measured at SMEAR-II mast at 16.8 m above ground with Pt100 sensor. From ATTO we use temperature measured at 81 m measured with CS215-L Digital Air Temperature and Relative Humidity Sensor, which is close to the particle measurements and has the best data coverage. For SMEAR-II, we have discarded measurements when the wind direction (hourly resolution) was between 120° and 140° to exclude the influence of the emissions from a nearby saw mill[53]. All data is averaged to hourly values to be comparable to the model output.

### Model and simulation descriptions

Key characteristics of the models are found in Table 1 and details are found in section S11.

### Model SOA yields

The models all have between 1 and three classes of oxidation products for BVOC. NorESM and EC-Earth both have two oxidation products which approximate extremely low volatility organic compounds (ELVOC) and low volatility organic compounds (LVOC). Both are treated as essentially non-volatile during condensation (condensation is calculated separately), but only ELVOC can participate in NPF and early growth. The molar (mass) yields for NorESM are as follows: IP + (OH/O$_3$/NO$_3$) → 5(12.33)%LVOC, MT + (OH/NO$_3$) → 15(18.5)%LVOC, O$_3$ + MT → 15(18.5)% ELVOC. The molar(mass) yields for EC-Earth are as follows: IP + OH → 0.97(3.3)%LVOC + 0.03(0.11)% ELVOC, IP +

O$_3$ → 0.99(3.37)% LVOC + 0.01(0.036)% ELVOC, MT + OH → 14(23.8)% LVOC + 1(1.8)% ELVOC and MT + O$_3$ → 10(17)% LVOC + 5(9.1)% ELVOC. UKESM has one oxidation product denoted LVOC which is treated as essentially non-volatile during condensation. The molar (mass) yields for UKESM are as follows: MT + (OH/O$_3$/NO$_3$) → 26(28.7)%LVOC. ECHAM-SALSA has a volatility basis set (VBS) parameterisation with 3 bins with saturation vapour concentrations at standard temperature and pressure (STP) of 0, 1, and 10 $\mu$g/m3 denoted as V$_0$, V$_1$, V$_{10}$, respectively in the following. Partitioning between the gas and particle phase is calculated by solving the condensation equations for each size bin. The molar (mass) yields are as follows: MT + (OH/O$_3$/NO$_3$) → 10(10)% V$_0$ + 3.7(3.7)% V$_1$ + 8.5(8.5)% V$_{10}$, IP + (OH/O$_3$/NO$_3$) → 2.95(5.9)%V$_1$ + 4.53(9.06)%V$_{10}$ for oxidants. See also Tables S4 and S5 for a full overview.

### Simulation description

We use the same setup for all models as far as possible, simulating the period from 2012 to and throughout 2018, using 2011 as a spin-up year. All models use nudging to ERA-Interim data[54] with a relaxation time of 6 hours. The nudging variables varies slightly, see details in section S11 (divergence, vorticity and surface pressure in EC-Earth and ECHAM-SALSA and horizontal winds in UKESM and horizontal winds and surface pressure in NorESM). We use historical emissions based on the Community Emissions Data System (CEDS) inventory as recommended for historical simulations in CMIP6[55] up until the end of 2014 and after this, we use SSP2-4.5 emissions[56]. Due to technical limitations, EC-Earth used SSP3-7.0 for the emissions.

### Comparing models and observations

We output hourly data from the models, with the exception of UKESM, for which we only had 3 hourly output available, and the cloud properties for EC-Earth, which can only be output in 3 hourly resolution. For both stations we use the grid cell covering the station. We use the bottom model level of the modelled atmosphere for SMEAR-II and the second level for ATTO for the analysis. The bottom layer typically covers approximately 100 metres above the surface, which covers the inlet heights at both sites, but since the inlet for the aerosol measurements at ATTO is at 60 m we use the second layer to account for possibly unrepresented boundary layer dynamics, for example that the measurements be outside of the nocturnal boundary layer (see diurnal variability for ATTO in Fig. S50). The choice of the level does not have a strong effect on the results (see Figs. S4 and S5). For the models where it is possible (UKESM and EC-Earth), we omit OA in the coarse mode when calculating the total OA because the measurements include only PM1. In NorESM, OA cannot contribute much to the coarse mode (POA is mainly emitted in PM1 and SOA will condense according to condensation sink which is dominated by PM1 particles), so total OA is an acceptable approximation for PM1. Similarly for ECHAM-SALSA, the partitioning of SOA is calculated by solving the condensation equation for all sizes. However, since the area-to-volume ratio is higher for PM1 than for the coarse mode, SOA will have a more significant contribution to PM1 that of the coarse mode.

### Regression analysis

For the regression lines in Fig. 2 we use orthogonal distance regression (ODR) (using the python package scipy and the scipy.odr.ODR[57]), for fitting a linear regression of the form $\ln(OA) = aT + b$. For the linear regression between OA mass concentration and number concentrations in Figs. 3 and S17 we again use ODR. Least-square regression is used for the logarithmic regression in Fig. 3 because ODR regression is known to struggle with non-linear regression. We present logarithmic regression only when for OA and $N_{50}$ and $N_{100}$ at SMEAR-II, because we did not observe the same buffering in the observations for the other regressions. The fitted parameters and their standard deviations are listed in Tabs. S1 and S2.

## Cloud properties

For the observations, we follow the procedure used in[8], and use level-3 MODIS-Aqua data (MYD08_D3, 6.1) daytime values for COT and $r_{eff}$. For the models we use the median daytime (9–16 UTC+2 for SMEAR-II and UTC-4 for ATTO) values for the analysis. The analysis is limited to liquid clouds (cloud top temperature over $-15°C$), and use only data where CWP is above $50\,gm^{-2}$, COT between 5 and 50 and $r_{eff}$ larger than $5\,\mu m$. This is to get the most reliable cloud observations. For the models, we limit CWP and COT in the same way, but we limit $r_{eff}$ only to $r_{eff} > 1\,\mu m$ because the modelled distribution of $r_{eff}$ is shifted towards much smaller values than the observations (see Figs. S32–S34). While the histogram over the observed $r_{eff}$ declines to practically 0 before the $5\,\mu m$ limit, limiting the models to this value would mean removing significant parts of the distribution which could introduce artificial effects into the analysis (see Figs. S34 and S34). The area used for the analysis of cloud properties is 60–66°N, 22–30°E for SMEAR-II and 8–1°S, 67–52°W for ATTO.

Finally, ice clouds are filtered/masked from the model output not by cloud top temperature (which was not available in the model output), but rather via using output indicating ice (see details in section S12). To produce the final plots, the bins of CWP are uniformly distributed between the lowest value (given that below $50\,gm^{-2}$ values are already filtered out) and the 95th percentile of the CWP for each data source (observations or model output). The position of the CWP bin on the x-axis is the median CWP within the bin. For ATTO, UKESM is limited to CWP below $800\,gm^{-2}$ due to outliers (see Fig. S33b).

## Estimating confidence intervals for changes in cloud properties

Figure 4 shows the difference in median COT (a, b) and cloud droplet effective radius (c, d) between high OA (above 67th percentile) days and low OA (below 33rd percentile) days and thus represents the sensitivity of the clouds to the OA perturbations in the model or observations. We choose to consider OA, rather than e.g. $N_{100}$ because the appropriate CCN proxy would vary depending on environmental conditions, particularly water vapour mixing ratio and updraft - i.e. information that is not available for us for all the data points. The confidence intervals represent the sampling uncertainty for the difference in the median values for high versus low concentrations are estimated by bootstrapping with 50,000 iterations and the percentile method. In other words, each group is sampled with replacement 50.000 times and for each pair of samples, the difference in median between high and low concentrations is calculated. The confidence interval's high and low limits are taken to be the 5th and 95th percentile of the resulting distribution. The median shows the original sample median (not the median of the bootstrap samples). The random retrieval for each grid cell in the level 3 MODIS data product used in the analysis are around 15%[58] and the temporally aggregated values used in this study will be considerably lower and is therefore not considered in the analysis.

## Comparison of total feedback strength in Fig. 5

To compare the overall effect of the chain of processes outlined in Fig. 1 in observations and models, we calculated the change in $r_{eff}$ and COT (shown in Figs. 5 and S6) with temperature by simply combining the results using $\frac{dX}{dT} = \frac{dOA}{dT} \cdot \frac{dX}{dOA}$, where $X$ is $r_{eff}$ or COT. We estimate $\frac{dOA}{dT}$ for each data source, i.e. a model or observations, by taking the best fit exponential function for the source, $OA = f(T)$, and calculating the $\frac{dOA}{dT} \approx \frac{\Delta OA}{\Delta T} \approx \frac{f(T_{med}+3)-f(T_{med})}{3}$, where $T_{med}$ is the median measured temperature for the seasons in question (15.6°C for SMEAR-II and 25.5°C for ATTO) and 3°C is the best estimate of the climate sensitivity[4]. We estimate the $\frac{dX}{dOA}$ by first estimating $\Delta X$ by averaging over the changes in $X$ between high and low OA in Fig. 4 and weighting by the number of days in each CWP bin. Then we use the difference between the median of the high and the low OA days as an estimate for $\Delta OA$, finally calculate

$\frac{dX}{dOA} \approx \frac{\Delta X}{\Delta OA}$. Note: (1) that for UKESM at ATTO, the correlation between temperature and organic aerosol is almost zero (see Table S2) and we therefore set the change in OA with temperature to zero for this model at this station. (2) For ECHAM-SALSA we use the fit where 2015/2016 are excluded because these years are clearly anomalous and not related to the biogenic emissions (see section S4).

## Data availability

The model data generated in this study have been deposited in the Bolin Center database at[59]. The SMEAR-II particle number size distribution data is available in the EBAS database at ebas-data.nilu.no. The SMEAR-II OA mass concentration data is available in the EBAS database at ebas-data.nilu.no. The temperature and wind measurement data from SMEAR-II is in the AVAA database at https://smear.avaa.csc.fi. The ATTO station measurement is available in the ATTO data portal at https://www.attodata.org. Source data are provided with this paper.

## Code availability

The analysis code is available at[60].

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

## Acknowledgements

We acknowledge funding from the following sources: The European Commission grants FORCeS no. 821205 (I.R., S.B., L.H., C.M., AV, T. Y., T.P., D.G.P.); INTEGRATE no. 867599 (I.R., S.B., L.H., C.M.), CRiceS No 101003826 (T.P.), FOCI no. 101056783 (I.R., M.K., T.P., T.M.), ATM-GTP no. 742206 (M.K.,T.P.), PyroTRACH no. 726165 (E.H.). Knut and Alice Wallenberg foundation no. 2017.0165 (C.M.), no. 2021.0169 (I.R.), no. 2021.0298 (I.R.) and no. 2022.0104 (C.M., I.R.). Academy of Finland no. 339885 (T.M.), no. 317390 (T.M., H.K.). Bundesministerium für Bildung und Forschung (BMBF) contracts 01LB1001A (C.P., P.A., C.Q.D.), 01LK1602B (C.P.), and 01LK2101B (C.P.). The Brazilian Ministério da Ciência, Tecnologia e Inovação (MCTI/FINEP) contract 01.11.01248.00 (P.A., C.Q.D., C.P.). The Conselho Nacional de Desenvolvimento Científico e Tecnológico (CNPq, Brazil) process 200723/2015-4 (P.A., C.Q.D., C.P.) and 169842/2017-7 (P.A.), 307530/2022-1 (C.Q.D.). The FAPESP (Fundação de Amparo à Pesquisa do Estado de São Paulo) grant no. 2017/17047-0 (P.A.) and no. 2020/15405-0 (B.B.M.). CAPES project grant no. 88887.368025/2019-00 (P.A.). ACCC Flagship funded by the Academy of Finland grant numbers 337549 (M.K., T.P.), 337550 (A.V., T.Y.). Academy professorship funded by the Academy of Finland grant no. 302958 (M.K., T.P.). Academy of Finland projects no. 334792 (M.K., T.P.), 325681 (M.K., T.P.), 347782 (M.K., T.P.), 328616 (M.K., T.P.), 347780 (M.K., T.P.). The Strategic Research Council (SRC) at the Academy of Finland no. 352431 (M.K., T.P.) and no. 335562 (H.K., T.M.). "Quantifying carbon sink, CarbonSink+ and their interaction with air quality" INAR project funded by Jane and Aatos Erkko Foundation (M.K., T.P.). "Gigacity" project funded by Wihuri foundation (M.K., T.P.). Natural Environment Research Council grant no. NE/S015396/1 (C.E.S.). Natural Environment Research Council grant no. NE/W001713/1 (D.G.P.). Svenska Forskningsrådet Formas grant no 2018-01745 (M.S., C.S.). NERC GW4+ award no. NE/L002434/1 (E.T.). For the operation of the ATTO site, we acknowledge the support by the Instituto Nacional de Pesquisas da Amazônia (INPA), the Amazon State University (UEA), the Large-Scale Biosphere-Atmosphere Experiment (LBA), FAPEAM, the Reserva de Desenvolvimento Sustentável do Uatumã (SDS/CEUC/RDS-Uatumã), the Max Planck Society, as well as all people involved in the technical, logistical, and scientific support of the ATTO project. We similarly acknowledge the people responsible for maintaining the long-term

measurements at SMEAR-II. We acknowledge University of Helsinki support via ACTRIS-HY. The simulations with NorESM and EC-Earth and the data analysis were by resources provided by the National Academic Infrastructure for Supercomputing in Sweden (NAISS) and the Swedish National Infrastructure for Computing (SNIC) at NSC partially funded by the Swedish Research Council through grant agreements no. 2022-06725 and no. 2018-05973 (projects 2022/2-1, 2022/1-1, 2022/6-272 and 2022/1-3). The simulations with LPJ-GUESS for the EC-Earth simulations were enabled by resources provided by LUNARC (LU 2021/2-114). The UKESM simulations were done with the aid of the Monsoon2 system, a collaborative facility supplied under the Joint Weather and Climate Research Programme, a strategic partnership between the Met Office and the Natural Environment Research Council. ECHAM-HAMMOZ is developed by a consortium composed of ETH Zürich, Max Planck Institut für Meteorologie, Forschungszentrum Jülich, University of Oxford, the Finnish Meteorological Institute, and the Leibniz Institute for Tropospheric Research and managed by the Center for Climate Systems Modelling (C2SM) at ETH Zürich. Dr. Alistair Sellar provided support for the configuration of the UKESM1 simulations performed as part of the AeroCom GCM Trajectory experiment on which these simulations are based. We also thank all the people responsible for the development of UKESM1. S.B. would like to acknowledge Diego Aliaga for valuable discussions and Tim Carlsen for advice on the satellite products.

## Author contributions

S.B.: Study design, data analysis, writing manuscript. T.Y.: Study design, data analysis, interpretation of results, helped write the manuscript (MS). T.M.: Study design, advice on analysis, interpretation of results, commenting MS. C.P.: Advice on analysis, field measurements, interpretation of results, commenting MS. E.H.: Performing simulations with ECHAM-SALSA, interpretation of the results, commenting MS. L.H.: Field measurements, interpretation of the results, commenting MS. C.M.: Study design, commenting on MS. P.A. Facilitation and design of long-term ATTO measurements. S.C. Field measurements and commenting MS. B.B.M.: Field measurements, interpretation of the results, commenting on MS. C.Q.D.: Field measurements, commenting on the MS. M.K.: Facilitation and design of long-term SMEAR II measurements, commenting MS. T.P.: Facilitation and design of long-term SMEAR II measurements, commenting MS. C.E.S.: Interpretation of the results, commenting MS. C.S.: Performed the simulations with EC-Earth, interpretation of results, commenting MS. L.N. Provided BVOC emissions from LPJ-Guess4.1 for EC-Earth simulations, commenting MS. M.S.: Configuration of the EC-Earth simulations, Interpretation of results, commenting MS. D.G.P.: Configuration of the UKESM simulations, interpretation of results, commenting MS. E.T.: Performing UKESM simulations and post-processing data, commenting MS. A.V.: Study design, interpretation of results, commenting MS. H.K.: Interpretation of results, commenting MS. I.R. Study design, data analysis, interpretation of results, helped write the MS.

## Funding

## Competing interests

The authors declare no competing interests.
