## [Peer Review File · Nature Communications]

Process-evaluation of forest aerosol-cloud-climate feedback shows clear evidence from observations and large uncertainty in modelsREVIEWER COMMENTS

Reviewer #1 (Remarks to the Author):

This manuscript is very interesting and useful. It uses compares the results of four Earth System Models (ESMs) with two long-term datasets of aerosol and cloud-relevant properties from two monitoring stations, one in the boreal and the second in a tropical forest as well as MODIS Aqua satellite observations, to investigate the efficacy of the ESMs in representing the aerosol-cloud-climate feedback over forests, involving biogenic volatile organics (BVOC) and secondary organic aerosol (SOA) formation during their oxidation.

The authors make the point that natural aerosols will be important drivers of aerosol climate impact in the future since anthropogenic emissions of aerosols and their precursors decrease due to regulations for air quality improvement. From their analysis, they find out significant errors in representing the feedback between BVOC/SOA/clouds and climate in the models that are limiting the accuracy of future climate predictions. This results points processes that need to be improved for climate prediction to be reliable.

The manuscript is well written, and the illustrations and discussion are appropriate and support the conclusions. I have a few comments to be addressed before the manuscript is accepted for publication.

1- The authors focus on the PM1 fraction of the organic aerosol since as they explain it is in the fine aerosol fraction that organics are expected to have the largest impact on clouds. They thus make a rigorous comparison between the model and observations based on PM1 (submicron) analysis. However, SOA mass can also be formed by condensation on larger particles or dissolution in cloud droplets, and this will modify the potential relationship between cloud number concentration and organic aerosol mass as a whole. While this is not an issue for the observations, the models are sharing the organic aerosol mass between the different aerosol sizes and consideration or not of the formation of SOA on coarse aerosol will affect the submicron SOA and the here shown results. I would like to have a comment on this issue and how far the model results presented here are affected by the size-resolved representation of SOA.

2- Please make clear the link between equation (1) for the feedback (dF/dT) and the equation given in Fig 5 caption ($dReff/dT$) also used to discuss the feedback. In Fig. 5, the reader was expecting to see the dF/dT instead of the change in the effective radius. In addition, the discussion that follows Fig 5 is focusing on the forcing but shows the change in the effective radius.

3- Line 36, please rephrase since the sentence might be misunderstood that the aerosols will warm the atmosphere in the future.

4- Discussion on Fig 2 for ATTO (lines 80-92). Since the models (other than ECHAM-SALSA and UKESM) similar to the observations do not have results for the high-temperature range, it is difficult to discuss

how they will behave in high temperatures.

5- Line 86 : models

6- Line 114: To what could you attribute the buffering in the impact of N50 and N100 for high OA ?

7- Lines 139-141: A reference for the updraft-limited and CCN-limited cases would be useful here. I could suggest the model intercomparison paper by Fanourgakis et al ACP <https://doi.org/10.5194/acp-19-8591-2019> but more can be found in the literature.

8- Figure 4: It is correct that EC-Earth and UKESM do not figure in the top panels? Are there data missing ? Please mention that in the caption.

9- Line 161: relationship

10- Line 181-182: please explain better.

11- Line 267: remove 'the n'

Reviewer #2 (Remarks to the Author):

This manuscript breaks down the BVOC-aerosol-cloud climate feedback loop in observations and models to underlying observable process level dependences that could potentially inform model understanding and development. Namely the authors use ground-based measurements from the SMEAR (boreal) and ATTO (tropical) sites to determine the dependence of organic aerosol (during periods when biogenics dominate OA mass) on temperature and then how the aerosol number size distribution and subsequent satellite-derived cloud properties vary with OA concentration. They highlight large inconsistencies in the magnitude of the BVOC-aerosol-cloud feedbacks between models analysed in this study and observations. While certain models represent various aspects of the feedback loop well particularly in the boreal regions, there clearly exists large and varied structural errors in the models that prohibit a correct simulation of BVOC effects on climate and how these might change in a future warmer climate. Capturing such natural aerosol ES feedback processes as this is a very important role for ESMs and inclusion of such levels of complexity is key for effective quantification of the magnitude of aerosol-cloud climate feedbacks and their response to climate change in particular with projected decreasing anthropogenic aerosol contributions. As such this study offers some key interesting results which highlights (a) the importance and magnitude of the feedback in the observations and relevance for climate and (b) limits in current ESMs ability to capture such feedbacks. I like the approach taken in terms of evaluating along the process chain using 2 unique datasets representing tropical and boreal regions. As such this has the potential to be an impactful and highly useful paper but I do have a number of concerns and questions around the methodology and conclusions which I go into more detail on below. I recommend these are addressed prior to being acceptable for publication in Nature Communications.

While there is clearly a very strong correlation between OA and Nx at SMEAR station, this is clearly not the case at ATTO, indeed the observations show a negative correlation in many cases while the models show a strong positive correlation (Table S2). I don't see any discussion of what drives the different relationship at the two sites (boreal vs tropical). Clearly background climate, vegetation types, temperature and cloud regimes is key. I think it would really strengthen such a paper to back up the regional analysis at the two sites with a broader discussion around what drives the differences.

Also, what drives the diversity in responses across models? Why does UKESM have a flat temperature dependence at ATTO (Fig 2) while at the same time $dr_{eff}/d[OA]$ is negative (Fig 5)? What role do the cloud microphysics and convection schemes and uncertainties therein play in cloud responses? In my view you should link the findings from the model process evaluation to some targeted model improvements. This would enable the significance of the method to inform model improvements to be established. For me it is not useful/impactful just to conclude the models have large structural uncertainties without pinpointing specific areas that require improvement. In some parts there is a small attempt at this for instance highlighting hygroscopicity issues in UKESM and in the Supplement when the yield is scaled in NorESM – it would be good to explore this type of application more, perhaps even in one model to demonstrate effectiveness of method/applicability of these observations for model constraint. Pulling this type of assessment together in the Discussion would be highly valuable.

I remain unconvinced based on the analysis presented here that aerosols are driving a significant, detectable response in the satellite derived cloud properties. While I am not an expert in satellite retrievals of cloud properties I understand there are considerable uncertainties in such retrievals particularly in tropical regions. I recognise the data has been filtered to restrict analysis to liquid cloud only I believe such uncertainties in the underlying retrieval products themselves should be clearly outlined. Looking at the distributions of cloud properties in Figures S18-20, it is difficult to determine even in the observations a meaningful difference between high and low OA cases. What is the statistical significance of the difference presented in Figure 4? I would argue that in particular for the COD differences in Figure 4 the uncertainties span 0 on the y-axis. Also what role do the systematic biases in the models wrt these properties, particularly at ATTO, play? In the modelling space the convection schemes will likely be a key determinant of the COD in these regions and many of these ESM still don't have a link between the aerosol and deep convection. Can the authors comment on the relative importance of such interactions for the tropical BVOC feedback?

Why are different regression functions used for SMEAR and ATTO (log function for the former and a linear function for the latter) – this further highlights the need for a better discussion on drivers of the regional differences in the process chain relationships in my view.

You present results for Feb-Apr for ATTO but your analysis in the Supplement shows there is little change in the Jan-May period, so why not use the full period? You state you want to have a period when there is “not strong seasonal change in cloud properties” (L60 Supplement) but there is clearly an increase in reff over the Feb-April period for instance. Can you clarify how you define “not strong seasonal change”

Overall the manuscript is generally well written and good use of appropriate reference but I find the Discussion and Conclusions section a bit weak. It doesn't tie the different threads of the analysis coherently enough together in my view nor link directly back to Figure 1. This thus weakens the potential

impact and significance of the results. So the messaging needs a lot of tidying and clarification. The Methods section is not written in as clear a way as it could be and the Supplement while extensive again could be improved in terms of writing, with more attention to detail required (in accurate figure captions etc) – see Minor Comments.

MINOR COMMENTS:

L45-46: please ensure you include the correct sign of the feedback when referencing values from the literature. You currently quote positive BVOC-climate feedbacks in all cases in your manuscript but this is not the case. Thornhill et al. for instance (your ref [15]) reports a negative feedback of $-0.09 \text{ W m}^{-2} \text{ K}^{-1}$ which doesn't agree with your reported value of $0.001 \text{ W m}^{-2} \text{ K}^{-2}$.

L54 “the its aerosol particle size distribution dynamics” – should this be “and”

Line 66 +1 (for some reason there's a gap in line numbering here): “long term in-situ data sets” – not sure 4 and 6 years of data can be classified as long term.

Eqn 1: but this is not how you actually calculate the feedback in Fig 5 you jump straight from $d[\text{OA}]/dT$ to $d[\text{cloud prop}]/d[\text{OA}]$ plus you don't actually calculate the full feedback, dF/dT in this analysis so I would reword this to say “ We analyse the components of the feedback which can be approximated as ...”

Figure 2/Section 1 of Results. The observed temperature range at ATTO is quite narrow (~5-6 degrees) compared to SMEARII where is spans at least 25degrees. How might this narrower range impact the observed regression?

The OA-temperature dependence in UKESM looks very odd, why is this?

L86: movels  models

L110 “on SOA formation” \diamond to SOA formation”

L117 these models aren't just “converting” mass to number, many aerosol processes influence the mass but not the number and vice versa. This phrase presents the issue as overly simplistic in my view.

L133 “it turns out not to be due”  This is not due to ..

L153 “is detectable” \diamond “is detectable in observations” . Although see my major comment around the weak observed relationship.

Figure 4: While you compute regressions for OA vs T and N_x vs OA , when presenting the change in cloud properties you bin according to the cloud liquid water path, why? Surely if aerosol is a key driver changes in these cloud properties you would see a clear relationship between reff vs OA and CoT vs OA?

L157 you should connect your pts 1-4 here directly back to your Figure 1 and components of the feedback loop. Also why no COT data for EC-Earth and UKESM

L182 I don't agree with your conclusion that you cannot rule out the strongest estimates (of feedback), is it not evident from Fig 5 that the NorESM feedback is too strong?

Table 1: UKESM: note iBVOC model is used for both IP and MT, this is correctly states in the model description in Section S9.

Methods general: a number of abbreviations aren't explained (ELVOC, LVOC, VBS etc)

L227 incomplete reaction / missing reaction

L230 Do you nudge only the winds or winds and temperature?

Remarks on Supplement:

Figs S1 and S2, captions incorrect SMEAR is not represented here.

Table S1, N_{100} v OA , linear regression for NorESM model repeated twice, I think this should be the log regression? Also why do you just apply a linear regression for N_{200} in contrast to the log regressions

used for N50 and N100?

L110 amazon  Amazon

S6.2/ L114 is there a section missing here?

Fig S22 and S31 please state the time resolution of the data points in these timeseries plots

Fig S37 how is the weighting applied?

L145-149 This paragraph is not clearly written overall. For example, "the vegetation will respond in terms of density to meteorological conditions" makes absolutely no sense to me.

L151: EC-Eearth  EC-Earth

L206 poorly worded sentence

L209/219/220 standard pressure was done assuming standard pressure ???

L211 what do you mean by cloud top cloud time?

Reviewer #3 (Remarks to the Author):

Overall, I think the paper is of interest and importance and can potentially be published, but requires revision of the presentation of the analysis and of how the conclusions are presented. I will give my reasoning below.

****Noteworthiness of the results:**

The manuscript explores a proposed feedback between the biosphere and climate. The proposed feedback is based on an increase in the emissions of organic aerosol precursors, which in turn leads to an increase in the organic aerosol mass, and also possibly the number of aerosol particles of a sufficient size to change the properties of clouds once they are formed. Several studies have been performed to study the magnitude of this effect, with varying results (as the authors note). Here, a comparison of several models is performed and the results are then compared with experimental data from two sites.

The work is of significance to the field and also original. The step-by-step exploration of the skill of the model in predicting the feedback process could be an important step forward if properly described. The results and findings, if properly justified, are certainly interesting and useful to the field, and the result is in my opinion noteworthy for at least two reasons:

- (1) the analysis shows that the proposed feedback is of a magnitude that should be explored when climate projections are made
- (2) the paper provides information for researchers, especially modellers, of specific areas that need updating

****Support of the conclusions and claims by the presented work, and possible flaws in the analysis:**

The approach to the analysis is fundamentally sound, and to me the technical details were of sufficient level. However, I think that the structure of the description of the analysis is not ideal, and should be improved. The changes that are required are to the structure of the presentation of the analysis, and might

also need some adjustment to the figures. I think the extent of the revision is significant, but new major analysis is not necessarily needed.

I think the main flaw in the data analysis is that the logic is somewhat confusing and difficult to follow. I will try to a) elaborate on how I understood it b) what I think is the problem, and c) suggest some improvements.

a) The biogenic emission - aerosol - cloud - temperature feedback is here analyzed by looking at different sub-processes of the loop. This is done by considering it as a chain of processes affecting radiative forcing F , for which the differential as a function of temperature reasonably close to the current state can be approximated with Eq. 1. The different terms of Eq. are analyzed, both from observational data, and data provided from ESM:s, and the responses of the models for each term are compared to responses from observational data. Based on this, the authors then make statements of the model's skill at each step of the feedback loop, and also conclude that the models showing a minimal feedback do not demonstrate sufficient agreement with observation to be considered reliable for this purpose.

b) The analysis then starts following Equation 1 piecewise by looking at each term. For the first two terms, the analysis is fairly straightforward - the dependence of OA on T is seen and can be clearly understood, and the basis for the dependence of CCN (in different cut-off size classes) on the OA mass is also described (with some minor questions still remaining).

When the analysis then reaches the third term (d cloud prop. / d CCN, according to both Fig 1 and Eq 1), the analysis then suddenly seems to change from this straightforward path. The cloud properties are not evaluated along the changes in CCN but rather OA. There is some discussion on CCN influence on cloud regimes, but the discussion focuses mostly on cloud water path (CWP), droplet effective radius (r_{eff}), and cloud optical thickness (COT). For the reader it does not become fully clear which of these is the dependent and which the independent variable that is being analyzed. For example, if the same approach than in parts (1) and (2) is followed, CWP would be the independent variable, and COP and r_{eff} the properties analyzed. How CWP is related to CCN is not really convincingly elaborated.

Instead, the analysis looks at $d(\text{cloud prop})/d\text{OA}$, and then arrives at quantification of $d(\text{cloud prop})/dT$. This is then used as the basis for the conclusions of the paper, along with some of the intermediate results.

A major point requiring clarification is why the last term - the radiative forcing - is left out of the analysis.

Thus, the problem as I see it is that the readers are proposed an analysis following Eq (1) and Fig (1), but they are then led along a more confusing route, and the deviations are not really explained. While I think that the analysis supports the results, the text does not really help in understanding this.

c) Here are my suggestions that the author might consider for the analysis:

Equation (1) makes sense, and it is understandable as an application of the chain rule, but it should be

justified a little better as it is the basis of the whole analysis. The different parts of Fig. (1) seem to connect with Eq (1) but there is not a full correspondence.

How I see this, is that actually the analysis aims to look at the process chain from T->cloud prop, and also analyses some of the intermediate steps (T->OA, OA->CCN), but does not focus on the steps CCN->cloud prop. The forcing part ($dF/d(\text{cloud prop})$) of Eq. 1 is completely ignored.

The authors would need to make a decision if they want to follow Eq(1) or not, and also structure the analysis along it. The deviations from equation (1) and fig(1) in the equations should be explained, and possibly also the connection between the equation and the figure could be enhanced by adding the terms to the figure.

It is also possible to not follow the equation to the detail, but to say that Eq 1 is the basic equation that underpins the idea of the analysis, but that then different sub-parts are analyzed in detail from the data (e.g. $dCOT/dOA$) to give insight on the process. It should, however, be made clear that the individual sensitivities are not analyzed in detail.

Once this decision is made, the description of the analysis could follow this plan, with each of the analyzed responses explained in the framework. It would be good if the reasoning behind Fig. (4) was explained more clearly, for example.

The authors can, of course, also choose an entirely different way of presenting the results. In any case, the presentation of Eq (1) as the basis of the analysis should either be updated or removed, as in current form this is not really followed.

**** Soundness and detail of methodology:**

The methodology is based on analysing two well-known, long-term datasets and a wealth of Earth System Model data. The data analysis methods are of sufficient quality, and choices made are generally well justified and also sufficient care has been taken to ensure that specific choices (such as the time period analyzed) are not unduly impacting results.

The methodology is presented in sufficient detail and the datasets used are in principle such that the work can likely be reproduced.

Other questions and comments:

Title, abstract, and e.g. l117: The authors use the term 'structural uncertainty in the model representation'. As one of the main claims is about structural uncertainty, maybe at least a reference to how such uncertainty is defined and evaluated (When is it large? How is it different from other uncertainties? How to address it) could be given for the reader.

Abstract: On line 26, the author state that they use 'natural variability as a proxy for perturbed states of

the climate'. This is not elaborated on in the paper at all. Either remove or explain this in more detail.

l56: I think that currently "begging the question" is used as the authors use it, but some people might consider this bad English (see e.g. <https://www.theguardian.com/media/mind-your-language/2010/may/24/begging-the-question-mind-your-language>)

l130: '...a similar pattern to the observed...' This seems very qualitative to me. Why is the pattern in Fig 4. significant, and how is the similarity evaluated? As this is the final and quite important step of the analysis, some more detail on the strength of the response was evaluated could be useful here. The same applies to line 136 ('...a reasonable response...'): what is reasonable? At which point would it become unreasonable?

l200: is the word 'again' needed here?

Reviewer #1 (Remarks to the Author):

This manuscript is very interesting and useful. It compares the results of four Earth System Models (ESMs) with two long-term datasets of aerosol and cloud-relevant properties from two monitoring stations, one in the boreal and the second in a tropical forest as well as MODIS Aqua satellite observations, to investigate the efficacy of the ESMs in representing the aerosol-cloud-climate feedback over forests, involving biogenic volatile organics (BVOC) and secondary organic aerosol (SOA) formation during their oxidation.

The authors make the point that natural aerosols will be important drivers of aerosol climate impact in the future since anthropogenic emissions of aerosols and their precursors decrease due to regulations for air quality improvement. From their analysis, they find out significant errors in representing the feedback between BVOC/SOA/clouds and climate in the models that are limiting the accuracy of future climate predictions. This results points processes that need to be improved for climate prediction to be reliable.

The manuscript is well written, and the illustrations and discussion are appropriate and support the conclusions. I have a few comments to be addressed before the manuscript is accepted for publication.

We would like to thank the reviewer for positive assessment and insightful suggestions! See below for point-by-point answers.

1- The authors focus on the PM₁ fraction of the organic aerosol since as they explain it is in the fine aerosol fraction that organics are expected to have the largest impact on clouds. They thus make a rigorous comparison between the model and observations based on PM₁ (submicron) analysis. However, SOA mass can also be formed by condensation on larger particles or dissolution in cloud droplets, and this will modify the potential relationship between cloud number concentration and organic aerosol mass as a whole. While this is not an issue for the observations, the models are sharing the organic aerosol mass between the different aerosol sizes and consideration or not of the formation of SOA on coarse aerosol will affect the submicron SOA and the here shown results. I would like to have a comment on this issue and how far the model results presented here are affected by the size-resolved representation of SOA.

For models where it was possible, we excluded coarse mode OA from the analysis (UKESM and EC-Earth). For NorESM, the contribution of OA to mass above 1 μm is limited, because it's only the SOA which could condense there and the surface area (and thus condensation sink) is largest in the accumulation mode. For ECHAM-SALSA, the partitioning of SOA is calculated by solving the condensation equation for all sizes. Since the area-to-volume ratio is higher for PM₁ than for the coarse mode, it is more sensitive to SOA. We have modified the text to include this (L356-361):

“For the models where it is possible (UKESM and EC-Earth), we omit OA in the coarse mode when calculating the total OA because the measurements include only PM₁. In NorESM, OA cannot contribute much to the coarse mode (POA is mainly emitted in PM₁ and SOA will condense according to condensation sink which is dominated by PM₁ particles), so total OA is an acceptable approximation for PM₁. Similarly for ECHAM-SALSA, the partitioning of SOA is calculated by solving the

condensation equation for all sizes. However, since the area-to-volume ratio is higher for PM1 than for the coarse mode, SOA will have a more significant contribution to PM1 than that of the coarse mode.”

2- Please make clear the link between equation (1) for the feedback (dF/dT) and the equation given in Fig 5 caption (dR_{eff}/dT) also used to discuss the feedback. In Fig. 5, the reader was expecting to see the dF/dT instead of the change in the effective radius. In addition, the discussion that follows Fig 5 is focusing on the forcing but shows the change in the effective radius.

We agree with this comment, and have modified the original equation. The section now reads:

“The total feedback is by definition the change in radiative forcing (F) with temperature (T) and can be decomposed as follows:

$$\frac{dF}{dT} = \frac{d(OA)}{dT} \cdot \frac{d(CCN)}{d(OA)} \cdot \frac{d(\text{cloud prop.})}{d(CCN)} \cdot \frac{dF}{d(\text{cloud prop.})}$$

where CCN is cloud condensation nuclei concentration, OA is the organic aerosol mass, “cloud prop.” refers to cloud properties. In this study, we target the terms in the feedback up until changes in cloud properties, i.e.

$$\begin{aligned} \frac{d(\text{cloud prop.})}{dT} &= \frac{d(OA)}{dT} \cdot \frac{d(CCN)}{d(OA)} \cdot \frac{d(\text{cloud prop.})}{d(CCN)} \\ &= \frac{d(OA)}{dT} \cdot \frac{d(\text{cloud prop.})}{d(OA)} \end{aligned}$$

The relationship between changes in cloud properties and forcing ($\frac{dF}{d(\text{cloud prop.})}$) is currently an active area of research in its own right [e.g. 3, 42], which we, therefore, leave outside of the scope of this study. Note that using present-day conditions to evaluate the feedback limits our analysis to the “pure” temperature feedback and excludes the potential effect of CO₂ fertilization on BVOC emissions, which has been shown to be a large contributor in some ESMs [see e.g. 43]. We use number concentrations of particles larger than 50 nm, 100 nm and 200 nm (N₅₀, N₁₀₀, N₂₀₀) to investigate the link between OA and CCN, “cloud prop.” is investigated by changes in cloud droplet effective radius and cloud optical thickness, and we evaluate the terms $\frac{d(CCN)}{d(OA)}$ and $\frac{d(\text{cloud prop.})}{d(OA)}$ both separately and combined, i.e. $\frac{d(\text{cloud prop.})}{d(OA)}$, to assess the combined effect. In our process-based evaluation, we combine the insights from relevant observational data sets with the unrivalled ability of ESMs to produce projections on the global scale.”

3- Line 36, please rephrase since the sentence might be misunderstood that the aerosols will warm the atmosphere in the future.

Good suggestion. The sentence now reads: “The importance of natural aerosols and the feedbacks associated with them may hence increase (again) as we move into a warmer future where air pollution mitigation is expected to give a cleaner atmosphere and thus a reduced aerosol cooling”

4- Discussion on Fig 2 for ATTO (lines 80-92). Since the models (other than ECHAM-SALSA and UKESM) similar to the observations do not have results for the high-temperature range, it is difficult to discuss how they will behave in high temperatures.

Yes, this is a good point, and it turns out that these anomalously high temperatures are from the years 2015/2016 for both UKESM and ECHAM-SALSA. For ECHAM-SALSA these two years seem to be dominated by something other than temperature suppressing the emissions (see new supplementary Figure S9 and S12-13). Given the anomaly in these years of the simulation data, we have now updated the main figure (Fig. 2, see below) to also include a regression and a distinction between all the data and excluding the 2015 and 2016 data. These show no substantial change for UKESM (still no correlation), and neither the observations, NorESM or EC-Earth. However, ECHAM-SALSA changes substantially and now resembles the observations quite well.

5- Line 86 : models

Thanks, corrected!

6- Line 114: To what could you attribute the buffering in the impact of N50 and N100 for high OA ?

Thanks for this question! In response to this comment and a question about the log fit from reviewer II we have added the following discussion to the text (L150-165):

“At SMEAR-II observations reveal a distinct buffering in the impact on N50 and N100 for high OA concentrations and the relationship is well captured by logarithmic

function, $N_x = a + b \ln(c + OA)$. This buffering is not seen in the same way at ATTO. A buffering could be expected for many different reasons: for one, with increased OA and thus number concentration, the organic vapors will condense onto more particles, thus limiting buffering the growth of each particle per increase in OA. Secondly, high loadings of OA could inhibit new particle formation (NPF) which will be suppressed by both due to increased condensations sink (reducing pre-cursor concentrations) and coagulation sink (reduces survival of particles to larger sizes) [e.g. 44]. Since SMEAR-II is known to have frequent NPF, even in summer [45], while ATTO is known to have very little [22], which could explain the discrepancy between the two stations. On the other hand, it could also be due to the aerosol concentrations (both number and mass) simply being too low at ATTO compared to SMEAR-II, so that the buffering would only be seen at higher concentrations. The buffering seems to be captured in all models except NorESM, but it is too strong in EC-Earth and ECHAM-SALSA and too weak in UKESM. None of the models have buffering at ATTO. The observed relationships between OA and number concentration for the two environments (SMEAR-II and ATTO) are almost identical if both are fitted with a linear regression (see Fig. S15 and Fig. S16). The models, however, predict a larger difference between the two sites, with the exception of UKESM. The consistency of the observed slope may of course be incidental, or it might be a symptom of some process constraining the size distribution dynamics in reality which is currently not well represented in models.”

7- Lines 139-141: A reference for the updraft-limited and CCN-limited cases would be useful here. I could suggest the model intercomparison paper by Fanourgakis et al ACP <https://doi.org/10.5194/acp-19-8591-2019> but more can be found in the literature.

Good suggestion, we included a reference to Fanourgakis et al., (2019) here.

8- Figure 4: It is correct that EC-Earth and UKESM do not figure in the top panels? Are there data missing ? Please mention that in the caption.

Good point, we added to the figure caption: “Note that the COT output was not available for EC-Earth and UKESM and these are therefore only shown in the lowermost panels.”

9- Line 161: relationship

Thanks! This was actually supposed to be “these relationships” because it’s for N50, N100 and N200. The sentence now reads “These relationships between OA mass concentration and particle number concentration affect the strength of the OA on clouds.”

10- Line 181-182: please explain better.

Thanks for noticing this. In response to this, and reviewer II’s comment on the same sentence, we have revised the sentence. It is evident that NorESM is too strong in the boreal zone. It is not, however, evident that it is too strong in the tropics and this is why we do not rule it out. We have clarified this as follows (L289-293):

“Overall, our study seems to rule out the lowest model estimates of the feedback (UKESM). Although the strongest model estimates (NorESM) is revealed to

overestimate the feedback in the boreal zone, the compensating error in the tropics makes it hard to completely rule out it out, especially considering the important role the tropics play in the global radiation budget.”

11- Line 267: remove ‘the n’

Thanks, corrected.

Reviewer #2 (Remarks to the Author):

This manuscript breaks down the BVOC-aerosol-cloud climate feedback loop in observations and models to underlying observable process level dependences that could potentially inform model understanding and development. Namely the authors use ground-based measurements from the SMEAR (boreal) and ATTO (tropical) sites to determine the dependence of organic aerosol (during periods when biogenics dominate OA mass) on temperature and then how the aerosol number size distribution and subsequent satellite-derived cloud properties vary with OA concentration. They highlight large inconsistencies in the magnitude of the BVOC-aerosol-cloud feedbacks between models analysed in this study and observations. While certain models represent various aspects of the feedback loop well particularly in the boreal regions, there clearly exists large and varied structural errors in the models that prohibit a correct simulation of BVOC effects on climate and how these might change in a future warmer climate.

Capturing such natural aerosol ES feedback processes as this is a very important role for ESMs and inclusion of such levels of complexity is key for effective quantification of the magnitude of aerosol-cloud climate feedbacks and their response to climate change in particular with projected decreasing anthropogenic aerosol contributions. As such this study offers some key interesting results which highlights (a) the importance and magnitude of the feedback in the observations and relevance for climate and (b) limits in current ESMs ability to capture such feedbacks. I like the approach taken in terms of evaluating along the process chain using 2 unique datasets representing tropical and boreal regions. As such this has the potential to be an impactful and highly useful paper but I do have a number of concerns and questions around the methodology and conclusions which I go into more detail on below. I recommend these are addressed prior to being acceptable for publication in Nature Communications.

We would like to thank the reviewer for these valuable and insightful comments. See below for point-by-point answers.

While there is clearly a very strong correlation between OA and Nx at SMEAR station, this is clearly not the case at ATTO, indeed the observations show a negative correlation in many cases while the models show a strong positive correlation (Table S2). I don't see any discussion of what drives the different relationship at the two sites (boreal vs tropical). Clearly background climate, vegetation types, temperature and cloud regimes is key. I think it would really strengthen such a paper to back up the regional analysis at the two sites with a broader discussion around what drives the differences.

Thanks for noticing this! Actually r^2 in Table 2 was supposed to be R^2 , i.e. the coefficient of determination, not the correlation (calculated using sklearn.metrics.r2_score). This is sometimes negative because the fitting is done minimizing orthogonal distance, not ordinary least square. R^2 is defined as $R^2 = 1 - \frac{SS_{res}}{SS_{tot}}$, where $SS_{res} = \sum_i (y_i - f(x_i))^2$ and $SS_{tot} = \sum_i (y_i - \bar{y})^2$. We will add the correlation coefficient to the table and furthermore correct to R^2 in the title.

In terms of discussion about the differences between the stations, we agree with the reviewer and have added the following to the introduction (L48-56):

“Boreal and tropical forests currently constitute about 27 and 45 % of global forested area, respectively [17]. These forest ecosystems are among the greatest sources of BVOCs emitted to the atmosphere [18] and the total global SOA burden [18, 19], and are therefore important drivers of potential BVOC feedbacks. Tropical forests are characterized by high diversity in tree species [20], while boreal forests have fewer species including a larger fraction of coniferous trees [17, 21]. This leads to a different spectrum of BVOCs emitted by these two ecosystems: the tropical BVOC emissions are dominated by isoprene (IP), while monoterpenes (MT) typically dominate the VOCs emitted from boreal forests [22, 23, 24, 25]. The differences between the tropical versus the boreal forest range from BVOC emission drivers [10, 26], the molecular spectra of the emitted species, oxidation chemistry [e.g. 27], the hydrological cycle and cloud regimes, and it is therefore vital to analyse both to understand the full impact of any feedback.”

Further, we’ve added some discussion about the differences between regressions at ATTO and SMEAR-II (L160-):

“The buffering seems to be captured in all models except NorESM, but it is too strong in EC-Earth and ECHAM-SALSA and too weak in UKESM. None of the models have buffering at ATTO. The observed relationships between OA and number concentration for the two environments (SMEAR-II and ATTO) are almost identical if both are fitted with a linear regression (see Fig. S15 and Fig. S16). The models, however, predict a larger difference between the two sites, with the exception of UKESM. The consistency of the observed slope may of course be incidental, or it might be a symptom of some process constraining the size distribution dynamics in reality which is currently not well represented in models.”

We have also added the following to the discussion/conclusion (L254):

“The models perform overall much more coherently with both each other and the observations in SMEAR-II compared to ATTO. This is perhaps not surprising: a long history of both forest and aerosol research performed at SMEAR-II means that the observations from this station have been used actively during the development of these models and components therein. It does, however, speak to the importance of both establishing long-term measurements in under-sampled parts of the atmosphere – especially in the southern hemisphere – and placing more weight on already existing measurements, like those from ATTO, in model development. It is, for example, possible that the fixed SOA yield approach used in these models works well enough in a boreal forest, but fails in a high isoprene environment like the tropics [see e.g. 27]. If so, this is a significant structural error in the models.”

Finally, we have added a section in the supplementary on the emissions in the models (see section S3) and references to this in the main in section (1) of the results (L110-116):

“Note that both ECHAM-SALSA and UKESM have anomalously high temperatures, and these are both for the years 2015 and 2016 (see discussion in section S3.2 and in particular Figs. S9 and S12). If these anomalous years are excluded, then ECHAM-SALSA is closer to the observations because these years have highly abnormal emissions (the β -term in the observations is 0.56 and ECHAM-SALSA has 0.20 with

2015/2016 and 0.46 without), while for UKESM the correlation stays similarly low as with these years (see Table S3)."

Also, what drives the diversity in responses across models? Why does UKESM have a flat temperature dependence at ATTO (Fig 2) while at the same time $dr_{eff}/d[OA]$ is negative (Fig 5)? What role do the cloud microphysics and convection schemes and uncertainties therein play in cloud responses? In my view you should link the findings from the model process evaluation to some targeted model improvements. This would enable the significance of the method to inform model improvements to be established. For me it is not useful/impactful just to conclude the models have large structural uncertainties without pinpointing specific areas that require improvement. In some parts there is a small attempt at this for instance highlighting hygroscopicity issues in UKESM and in the Supplement when the yield is scaled in NorESM – it would be good to explore this type of application more, perhaps even in one model to demonstrate effectiveness of method/applicability of these observations for model constraint. Pulling this type of assessment together in the Discussion would be highly valuable.

This is a very good point, and we have expanded on the discussion in the model differences, in particular with respect to the steps between emissions and aerosol formation (see supplementary section S3 "Modelled emissions"). We are already planning to investigate model perturbations with the same methodology, but it is unfortunately outside of the scope of the current work.

“Why does UKESM have a flat temperature dependence at ATTO (Fig 2) while at the same time $dr_{eff}/d[OA]$ is negative (Fig 5)?”:

The temperature dependency of OA is not necessarily related to the impact of high/low OA on cloud properties. One is governed by the sources of organic aerosols in the model and the other is governed by how these aerosols are activated into cloud droplets. . While there is no temperature dependence for OA at ATTO in UKESM, there is some variability in OA concentration, allowing for determining dr_{eff}/dt from the difference in r_{eff} between high and low OA as defined here. However, this does not describe the feedback studied here, as the variability in OA is driven by other processes than temperature dependence.

“What role do the cloud microphysics and convection schemes and uncertainties therein play in cloud responses?”

We have improved the discussion about the cloud results and the text now uses the same analysis for high versus low N100 and N50 to pin point clearer the origin of the differences.

In terms of cloud microphysics, all the models use (Abdul-Razzak and Ghan, 2000) or the sectional version (Abdul-Razzak and Ghan, 2002) for the activation. However, UKESM and EC-Earth both have an updraft pdf approach and the other two models don't. In terms of cloud-microphysics, NorESM and ECHAM-SALSA both have double moment schemes, which should improve the representation of cloud aerosol interactions in general. One might speculate that the single moment scheme in UKESM is partially responsible for the narrow distribution of cloud droplet effective radius (see supplementary section S6), but the same is not true for EC-Earth for example. Another factor that could contribute in UKESM, is the fairly high and narrow distribution of aerosols in UKESM, especially at ATTO.

Overall, it is a very interesting question how the cloud microphysics schemes or convection schemes influence the results, and we feel like we have expanded on the issue in the paper now. However, we believe that to conclude on the speculations above would be outside the scope of the current study and would therefore refrain from adding too much speculation to the paper.

We have, however, added information about the warm cloud microphysics schemes in the models into table 1 in the method section now.

“Pulling this type of assessment together in the Discussion would be highly valuable.”

We’ve added the following paragraph to the discussion (L262-):

“The models perform overall much more coherently with both each other and the observations in SMEAR-II compared to ATTO. This is perhaps not surprising: a long history of both forest and aerosol research performed at SMEAR-II means that the observations from this station have been used actively during the development of these models and components therein. It does, however, speak to the importance of both establishing long-term measurements in under-sampled parts of the atmosphere – especially in the southern hemisphere – and placing more weight on already existing measurements, like those from ATTO, in model development. It is, for example, possible that the fixed SOA yield approach used in these models works well enough in a boreal forest, but fails in a high isoprene environment like the tropics [see e.g. 27]. If so, this is a significant structural error in the models. In this study, two of the models have interactive oxidant chemistry, UKESM and EC-Earth. This is in contrast to NorESM and ECHAM-SALSA, where oxidant concentrations are read from file and cannot be affected (e.g. depleted) by changes in the BVOC emissions. It is interesting that these models both have too low a slope or a too weak relationship between temperature and OA mass (Fig. 2). This is especially true for UKESM, where the temperature dependency of monoterpene emissions is very similar to NorESM and ECHAM-SALSA (they all use MEGAN or pre-runners [10, 48]), meaning that the dependency of OA mass on temperature is lost during the oxidation process. A recent study by [49] using UKESM emphasises that oxidant chemistry can play a major role in the total feedback, also for the cloud-aerosol interactions. While the oxidant chemistry response is clearly missing in models like NorESM and ECHAM-SALSA that read oxidants concentrations from file, the fact that the models in this study with interactive chemistry (EC-Earth and UKESM) agree poorly with the observations for the temperature to OA relationship might encourage further improvements of the oxidant chemistry in the tropical forest regions in the models.”

I remain unconvinced based on the analysis presented here that aerosols are driving a significant, detectable response in the satellite derived cloud properties. While I am not an expert in satellite retrievals of cloud properties I understand there are considerable uncertainties in such retrievals particularly in tropical regions. I recognise the data has been filtered to restrict analysis to liquid cloud only I believe such uncertainties in the underlying retrieval products themselves should be clearly outlined. Looking at the distributions of cloud properties in Figures S18-20, it is difficult to determine even in the observations a meaningful difference between high and low OA cases. What is the statistical significance of the difference presented in Figure 4? I would argue that in particular for the COD differences in

Figure 4 the uncertainties span 0 on the y-axis. Also what role do the systematic biases in the models wrt these properties, particularly at ATTO, play? In the modelling space the convection schemes will likely be a key determinant of the COD in these regions and many of these ESM still don't have a link between the aerosol and deep convection. Can the authors comment on the relative importance of such interactions for the tropical BVOC feedback?

Thanks for this comment, it is a good point.

The sampling uncertainty is taken into account through the 5-95th percentiles shown in the cloud properties plots. If this spans the zero line, the results are not significant with 5 percent significance level (10 percent if you consider it two-tailed). This is why we sometimes mention it is not significant. It does not span the 0 for most of the bins in the observations though.

For the retrieval error, we have now added the following to the method section (L395):

“The random retrieval for each grid cell in the level 3 MOSIS data product used in the analysis are around 15 % [87] and the temporally aggregated values used in this study will be considerably lower and is therefore not considered in the analysis.”

Similar approaches have been taken both in e.g. (Yli-Juuti et al., 2021) on which we base the analysis here, but also e.g. in (McCoy et al., 2020). Note also that we have filtered the cloud data for liquid water clouds so that deep convection will be excluded due to cloud top ice, so the deep convection scheme will not play a large role here.

It is a good point to link these cloud responses back to general biases in the models. We have expanded upon this discussion now (L201):

“Note that the distribution of CWP in UKESM is quite heavily skewed towards lower values, meaning that the two smallest bins showing a significant response in 4 for SMEAR-II actually constitute most of the data. EC-Earth has a similar lack of response in the higher CWP bins for N100 and N50 as for OA. This limits the impact of the feedback and is not in accordance with the observations, especially for the middle values of CWP. For ATTO, all the models underestimate the change in reff with elevated OA concentrations, possibly with the exception of ECHAM-SALSA. On the other hand, for COT, NorESM is very similar to the observations, though with some non-significant overestimation for the middle CWP bins. ECHAM-SALSA overestimates the response for all CWP bins and is more than a factor of two too high for all bins above 170 gm⁻². Note that both models show inconsistent responses in reff compared to COT, while the observations mostly show the response to be mirrored between the two. The low response of the modelled reff for all the models in ATTO might be related the droplet sizes being smaller in the models (see Fig. S32b), making the clouds less susceptible to perturbation (updraft- rather than CCN-limited, [see e.g. 32]). This is consistent with the response of the cloud properties to high versus low N50 and N100, which is similarly low as for OA, and also the fact that the models overestimate aerosol concentrations at ATTO. In NorESM in particular, the extreme overestimation of both OA and number of particles in the region (Figs. 2 and 3), likely lead the cloud regime to be updraft- rather than CCN-limited. Note that this is the opposite picture to SMEAR-II where most models were too sensitive to changes in N50 and N100.”

Why are different regression functions used for SMEAR and ATTO (log function for the former and a linear function for the latter) – this further highlights the need for a better discussion on drivers of the regional differences in the process chain relationships in my view.

In general we have used the regression that best fit the measurements and at SMEAR for OA to N50 and N100, we found a good fit for a logarithmic function due to the buffering at high OA concentrations. We agree that this should be discussed more and have added the following discussion to the text (L150-165):

“At SMEAR-II observations reveal a distinct buffering in the impact on N50 and N100 for high OA concentrations and the relationship is well captured by logarithmic function, $N_x = a + b \ln(c + OA)$. This buffering is not seen in the same way at ATTO. A buffering could be expected for many different reasons: for one, with increased OA and thus number concentration, the organic vapors will condense onto more particles, thus limiting buffering the growth of each particle per increase in OA. Secondly, high loadings of OA could inhibit new particle formation (NPF) which will be suppressed by both due to increased condensations sink (reducing pre-cursor concentrations) and coagulation sink (reduces survival of particles to larger sizes) [e.g. 44]. Since SMEAR-II is known to have frequent NPF, even in summer [45], while ATTO is known to have very little [22], which could explain the discrepancy between the two stations. On the other hand, it could also be due to the aerosol concentrations (both number and mass) simply being too low at ATTO compared to SMEAR-II, so that the buffering would only be seen at higher concentrations. The buffering seems to be captured in all models except NorESM, but it is too strong in EC-Earth and ECHAM-SALSA and too weak in UKESM. None of the models have buffering at ATTO. The observed relationships between OA and number concentration for the two environments (SMEAR-II and ATTO) are almost identical if both are fitted with a linear regression (see Fig. S15 and Fig. S16). The models, however, predict a larger difference between the two sites, with the exception of UKESM. The consistency of the observed slope may of course be incidental, or it might be a symptom of some process constraining the size distribution dynamics in reality which is currently not well represented in models.”

You present results for Feb-Apr for ATTO but your analysis in the Supplement shows there is little change in the Jan-May period, so why not use the full period? You state you want to have a period when there is “not strong seasonal change in cloud properties” (L60 Supplement) but there is clearly an increase in reff over the Feb-April period for instance. Can you clarify how you define “not strong seasonal change”

Yes, this is a very reasonable question. We excluded January because it is not the wet season and there is still quite some influence from long range transport. This could be seen from analysis of BC which is not part of the study. The change in reff seen in between February and April is indeed there, and we’ve added the following sentence to highlight this (L300):

“(1) biogenic SOA is expected to dominate the OA (wet season) [34] and (2) the influence of seasonal changes in cloud properties and OA are weak (see Fig. S16). Note that there is still an increase in reff over Feb–Apr.”

We could have used February to May, but did not want to extend the period over too many months to minimize the seasonal impact. This leaves March to May which we include in the supplementary along with the other possible choices.

Overall the manuscript is generally well written and good use of appropriate reference but I find the Discussion and Conclusions section a bit weak. It doesn't tie the different threads of the analysis coherently enough together in my view nor link directly back to Figure 1. This thus weakens the potential impact and significance of the results. So the messaging needs a lot of tidying and clarification. The Methods section is not written in as clear a way as it could be and the Supplement while extensive again could be improved in terms of writing, with more attention to detail required (in accurate figure captions etc) – see Minor Comments.

Thank you for this! We have expanded quite a lot on the Discussion and Conclusion section now in addition to improving readability in the Method and Supplement.

MINOR COMMENTS:

L45-46: please ensure you include the correct sign of the feedback when referencing values from the literature. You currently quote positive BVOC-climate feedbacks in all cases in your manuscript but this is not the case. Thornhill et al. for instance (your ref [15]) reports a negative feedback of $-0.09 \text{ W m}^{-2} \text{ K}^{-1}$ which doesn't agree with your reported value of $0.001 \text{ Wm}^{-2} \text{ K}^{-2}$.

Thanks for noticing. We've added a minus signs for the mentions of the negative feedback estimates. The 0.001 value refers to the UKESM feedback in the Thornhill study (Table 9). We see how this is confusing and have rephrased it to “[...], to completely negligible ($0.001 \text{ Wm}^{-2}\text{K}^{-1}$ for UKESM in [15, table 9])”.

L54 “the its aerosol particle size distribution dynamics” – should this be “and”

Thanks, corrected.

Line 66 +1 (for some reason there's a gap in line numbering here): “long term in-situ data sets” – not sure 4 and 6 years of data can be classified as long term.

We've followed (Yli-Juuti et al., 2021) here, but we have added a qualifier and now it says “emerging long term in-situ data sets”.

Eqn 1: but this is not how you actually calculate the feedback in Fig 5 you jump straight from $d[\text{OA}]/dT$ to $d[\text{cloud prop}]/d[\text{OA}]$ plus you don't actually calculate the full feedback, dF/dT in this analysis so I would reword this to say “ We analyse the components of the feedback which can be approximated as ...”

This is a good point which reviewer 3 also made. We've made changes to this and the full description can be seen in the replies to reviewer 3 below.

Figure 2/Section 1 of Results. The observed temperature range at ATTO is quite narrow (~5-6 degrees) compared to SMEARII where is spans at least 25degrees. How might this narrower

range impact the observed regression?

The OA-temperature dependence in UKESM looks very odd, why is this?

Good point, we've added a sentence on the temperature range to the manuscript now.

Concerning the UKESM response: We've added a new section in the supplementary about the emissions in the models (as mentioned above) and from this it would seem that the emissions of monoterpene (the only SOA precursor in UKESM) have a reasonable temperature dependency (the same as the other models), but the OA does not. This indicates that the error originates from the oxidation process. We've added a paragraph in the discussion about this (L262):

"In this study, two of the models have interactive oxidant chemistry, UKESM and EC-Earth. This is in contrast to NorESM and ECHAM-SALSA, where oxidant concentrations are read from file and cannot be affected (e.g. depleted) by changes in the BVOC emissions. It is interesting that these models both have too low a slope or a too weak relationship between temperature and OA mass (Fig. 2). This is especially true for UKESM, where the temperature dependency of monoterpene emissions is very similar to NorESM and ECHAM-SALSA (they all use MEGAN or pre-runners [10, 48]), meaning that the dependency of OA mass on temperature is lost during the oxidation process. A recent study by [49] using UKESM emphasises that oxidant chemistry can play a major role in the total feedback, also for the cloud-aerosol interactions. While the oxidant chemistry response is clearly missing in models like NorESM and ECHAM-SALSA that read oxidants concentrations from file, the fact that the models in this study with interactive chemistry (EC-Earth and UKESM) agree poorly with the observations for the temperature to OA relationship might encourage further improvements of the oxidant chemistry in the tropical forest regions in the models."

L86: movels  models

Thanks, corrected.

L110 "on SOA formation" \diamond to SOA formation"

Thanks, corrected.

L117 these models aren't just "converting" mass to number, many aerosol processes influence the mass but not the number and vice versa. This phrase presents the issue as overly simplistic in my view.

Yes, this is a valid point. We've updated it to: "Overall, these results reveal a large uncertainty in the modelled processes of SOA formation and highlight the importance of adequately capturing aerosol size distribution dynamics."

L133 "it turns out not to be due"  This is not due to ..

Thanks, corrected!

L153 “is detectable” \diamond “is detectable in observations” . Although see my major comment around the weak observed relationship.

Corrected accordingly.

Figure 4: While you compute regressions for OA vs T and Nx vs OA , when presenting the change in cloud properties you bin according to the cloud liquid water path, why? Surely if aerosol is a key driver changes in these cloud properties you would see a clear relationship between reff vs OA and CoT vs OA?

The cloud effect on clouds is binned by cloud water path to constrain the types of clouds as the meteorological conditions are quite different when you have cumulus, stratocumulus or stratus clouds. There is naturally a relationship between CWP and COT/CER (more water means thicker cloud, see also Yli-Juuti et al (2021)), and since we are here targeting the effect of the aerosols, it makes sense to bin by CWP to avoid this confounding factor. See also Yli-Juuti et al (2021) for more background on the methodology, which was developed there. We've also added the following clarification at the start of section 3 (L169):

“We here analyse the modelled and observed impact on cloud properties, as represented by cloud optical thickness and cloud droplet effective radius. By binning by cloud water path (CWP), we constrain the impact of different cloud regimes/types on our analysis and also effectively constrain it to mainly the cloud albedo effect (or the first indirect effect), leaving other aerosol-cloud interactions outside the scope of this study [3]. We present the change in cloud properties as a result of changes in OA, not CCN. This is because the activation diameter may vary both within and between models and reality, and to evaluate the feedback strength, it is easier to follow the signal as outlined in equation 3 by considering $d(\text{cloud props.})/d(\text{OA})$ rather than $d(\text{cloud props.})/d(\text{CCN})$. See the supplementary, section S6.2, for the same figure as 4 but with high versus low N100 and N50.”

L157 you should connect your pts 1-4 here directly back to your Figure 1 and components of the feedback loop. Also why no COT data for EC-Earth and UKESM

Yes, good point. Again, reviewer 3 discussed the structure in some detail, and the updates on this is found there. For EC-Earth and UKESM we did not have COT output, and this has now been noted in the figure caption “Note that the COT output was not available for EC-Earth and UKESM and these are therefore only shown in the lowermost panels.”

L182 I don't agree with your conclusion that you cannot rule out the strongest estimates (of feedback), is it not evident from Fig 5 that the NorESM feedback is too strong?

It is evident that NorESM is too strong in the boreal zone. It is not, however, evident that it is too strong in the tropics and this is why we do not rule it out. We have clarified this as follows (L289-293):

“Overall, our study seems to rule out the lowest model estimates of the feedback (UKESM). Although the strongest model estimates (NorESM) is revealed to overestimate the feedback in the boreal zone, the compensating error in the tropics makes it hard to completely rule out it out, especially considering the important role the tropics play in the global radiation budget.”

Table 1: UKESM: note iBVOC model is used for both IP and MT, this is correctly states in the model description in Section S9.

Methods general: a number of abbreviations aren't explained (ELVOC, LVOC, VBS etc)

Thanks, the abbreviations have now been specified.

About iBVOC: thanks, the misunderstanding was due to the monoterpene emissions in iBVOC using a parameterisation from Guenther et al., (1995). This has now been corrected in the table which now says: “iBVOC (Pacifco et al., 2012)(Pacifco et al., 2015), IP: (Pacifco et al., 2011), MT: (Guenther et al., 1995)”

L227 incomplete reaction / missing reaction

Yes, good point! It's been corrected now.

L230 Do you nudge only the winds or winds and temperature?

Thanks for noticing that this information was not sufficiently provided. We've now added the specific nudging for each model to the supplementary (section 10), and we've also added the following to the method section

“[...] 6 hours. The nudging variables varies slightly, see details in section 10 (divergence, vorticity and surface pressure in EC-Earth and ECHAM-SALSA and horizontal winds in UKESM and horizontal winds and surface pressure in NorESM).”

Remarks on Supplement:

Figs S1 and S2, captions incorrect SMEAR is not represented here.

Thanks, these are now corrected.

Table S1, N100 v OA , linear regression for NorESM model repeated twice, I think this should be the log regression? Also why do you just apply a linear regression for N200 in contrast to the log regressions used for N50 and N100?

Thanks for noticing this! We only applied a log regression when the observational data suggested that it would be a better fit than a linear regression. We have now added a sentence in the method section to emphasise this.

L110 amazone  Amazon

Thanks, corrected.

S6.2/ L114 is there a section missing here?

Thanks, this was an error left over from an earlier version. We removed the caption.

Fig S22 and S31 please state the time resolution of the data points in these timeseries plots

Thanks, this has been added now. They are all in hourly (observations and all models except UKESM) and 3 hourly resolution (UKESM).

Fig S37 how is the weighting applied?

We apply no weighting here, which entails that years with more data coverage get a stronger impact on the total mean.

L145-149 This paragraph is not clearly written overall. For example, “the vegetation will respond in terms of density to meteorological conditions” makes absolutely no sense to me.

This is a fair comment, and we corrected it to “This means that the vegetation is allowed to respond to meteorological conditions, soil moisture, nutrient availability and so on by growing more or less dense (leaf area index can change), but that the distribution (the land area covered by each vegetation type) of the vegetation remains set (not dynamic vegetation).”

L151: EC-Eearth  EC-Earth

Corrected.

L206 poorly worded sentence

Indeed, it's been reworded to “We use only CWP values below 800gm⁻² at ATTO because the distribution for UKESM has an unrealistically long tail”

L209/219/220 standard pressure was done assuming standard pressure ???

Sorry, it's now corrected to “[...] was done assuming standard pressure, since we do not have ambient pressure as a model output.”

L211 what do you mean by cloud top cloud time?

Cloud time is a measure in the model giving how much of the time step or time period output that there was a cloud there. Cloud top cloud time is this value for the cloud top specific output. Since in this context, it is synonymous to cloud fraction, we have changed it to cloud fraction since it is easier to understand.

Reviewer #3 (Remarks to the Author):

Overall, I think the paper is of interest and importance and can potentially be published, but requires revision of the presentation of the analysis and of how the conclusions are presented. I will give my reasoning below.

We would like to thank the reviewer for taking the time to give these good suggestions for the presentation of the results. These comments are highly appreciated and we agree that the presentation needs this sort of clarification. We've provided point-by-point replies below.

****Noteworthiness of the results:**

The manuscript explores a proposed feedback between the biosphere and climate. The proposed feedback is based on an increase in the emissions of organic aerosol precursors, which in turn leads to an increase in the organic aerosol mass, and also possibly the number of aerosol particles of a sufficient size to change the properties of clouds once they are formed. Several studies have been performed to study the magnitude of this effect, with varying results (as the authors note). Here, a comparison of several models is performed and the results are then compared with experimental data from two sites.

The work is of significance to the field and also original. The step-by-step exploration of the skill of the model in predicting the feedback process could be an important step forward if properly described. The results and findings, if properly justified, are certainly interesting and useful to the field, and the result is in my opinion noteworthy for at least two reasons:

- (1) the analysis shows that the proposed feedback is of a magnitude that should be explored when climate projections are made
- (2) the paper provides information for researchers, especially modellers, of specific areas that need updating

****Support of the conclusions and claims by the presented work, and possible flaws in the analysis:**

The approach to the analysis is fundamentally sound, and to me the technical details were of sufficient level. However, I think that the structure of the description of the analysis is not ideal, and should be improved. The changes that are required are to the structure of the presentation of the analysis, and might also need some adjustment to the figures. I think the extent of the revision is significant, but new major analysis is not necessarily needed.

I think the main flaw in the data analysis is that the logic is somewhat confusing and difficult to follow. I will try to a) elaborate on how I understood it b) what I think is the problem, and c) suggest some improvements.

a) The biogenic emission - aerosol - cloud - temperature feedback is here analyzed by looking at different sub-processes of the loop. This is done by considering it as a chain of processes affecting radiative forcing F , for which the differential as a function of temperature reasonably close to the current state can be approximated with Eq. 1. The different terms of Eq. are analyzed, both from observational data, and data provided from ESM:s, and the responses of the models for each term are compared to responses from observational data. Based on this, the authors then make statements of the model's skill at each step of the

feedback loop, and also conclude that the models showing a minimal feedback do not demonstrate sufficient agreement with observation to be considered reliable for this purpose.

b) The analysis then starts following Equation 1 piecewise by looking at each term. For the first two terms, the analysis is fairly straightforward - the dependence of OA on T is seen and can be clearly understood, and the basis for the dependence of CCN (in different cut-off size classes) on the OA mass is also described (with some minor questions still remaining).

When the analysis then reaches the third term ($d \text{ cloud prop.} / d \text{ CCN}$, according to both Fig 1 and Eq 1), the analysis then suddenly seems to change from this straightforward path. The cloud properties are not evaluated along the changes in CCN but rather OA. There is some discussion on CCN influence on cloud regimes, but the discussion focuses mostly on cloud water path (CWP), droplet effective radius (r_{eff}), and cloud optical thickness (COT). For the reader it does not become fully clear which of these is the dependent and which the independent variable that is being analyzed. For example, if the same approach than in parts (1) and (2) is followed, CWP would be the independent variable, and COP and r_{eff} the properties analyzed. How CWP is related to CCN is not really convincingly elaborated.

This is a good point. Our approach essentially only diagnoses changes to cloud albedo, or the first indirect effect. We've followed Yli-Juuti et al., (2021) in this regard, but we have now also added a clarification at the start of section 3 (see below). We thus bin by CWP to approximately constrain our comparison to the same cloud types or regimes, but do not take into account possible impacts on lifetime of clouds or how a cloud with longer lifetime could again feed back onto the COT response (would a longer living cloud be thinner by the end of its lifetime?). We also do not consider changes to cloud fraction for example, as we consider only in-cloud properties in the study.

Clarification at the start of section 3 (L169-):

"We here analyse the modelled and observed impact on cloud properties, as represented by cloud optical thickness and cloud droplet effective radius. By binning by cloud water path (CWP), we constrain the impact of different cloud regimes/types on our analysis and also effectively constrain it to mainly the cloud albedo effect (or the first indirect effect), leaving other aerosol-cloud interactions outside the scope of this study [3]. We present the change in cloud properties as a result of changes in OA, not CCN. This is because the activation diameter may vary both within and between models and reality, and to evaluate the feedback strength, it is easier to follow the signal as outlined in equation 3 by considering $d(\text{cloud props.})/d(\text{OA})$ rather than $d(\text{cloud props.})/d(\text{CCN})$. See the supplementary, section S6.2, for the same figure as 4 but with high versus low N100 and N50."

Instead, the analysis looks at $d(\text{cloud prop})/d\text{OA}$, and then arrives at quantification of $d(\text{cloud prop})/dT$. This is then used as the basis for the conclusions of the paper, along with some of the intermediate results.

A major point requiring clarification is why the last term - the radiative forcing - is left out of the analysis.

This is a good point and we have now clarified that we are addressing the $d(\text{cloud prop.})$ and leaving the $dF/d(\text{cloud prop.})$ out of the study. The introduction now contains this (L85):

“The total feedback is by definition the change in radiative forcing (F) with temperature (T) and can be decomposed as follows:

$$\frac{dF}{dT} = \frac{d(OA)}{dT} \cdot \frac{d(CCN)}{d(OA)} \cdot \frac{d(\text{cloud prop.})}{d(CCN)} \cdot \frac{dF}{d(\text{cloud prop.})}$$

where CCN is cloud condensation nuclei concentration, OA is the organic aerosol mass, “cloud prop.” refers to cloud properties. In this study, we target the terms in the feedback up until changes in cloud properties, i.e.

$$\begin{aligned} \frac{d(\text{cloud prop.})}{dT} &= \frac{d(OA)}{dT} \cdot \frac{d(CCN)}{d(OA)} \cdot \frac{d(\text{cloud prop.})}{d(CCN)} \\ &= \frac{d(OA)}{dT} \cdot \frac{d(\text{cloud prop.})}{d(OA)} \end{aligned}$$

The relationship between changes in cloud properties and forcing ($\frac{dF}{d(\text{cloud prop.})}$) is currently an active area of research in its own right [e.g. 3, 42], which we, therefore, leave outside of the scope of this study. Note that using present-day conditions to evaluate the feedback limits our analysis to the “pure” temperature feedback and excludes the potential effect of CO₂ fertilization on BVOC emissions, which has been shown to be a large contributor in some ESMs [see e.g. 43]. We use number concentrations of particles larger than 50 nm, 100 nm and 200 nm (N₅₀, N₁₀₀, N₂₀₀) to investigate the link between OA and CCN, “cloud prop.” is investigated by changes in cloud droplet effective radius and cloud optical thickness, and we evaluate the terms $\frac{d(CCN)}{d(OA)}$ and $\frac{d(\text{cloud prop.})}{d(OA)}$ both separately and combined, i.e. $\frac{d(\text{cloud prop.})}{d(OA)}$, to assess the combined effect. In our process-based evaluation, we combine the insights from relevant observational data sets with the unrivalled ability of ESMs to produce projections on the global scale.”

Thus, the problem as I see it is that the readers are proposed an analysis following Eq (1) and Fig (1), but they are then led along a more confusing route, and the deviations are not really explained. While I think that the analysis supports the results, the text does not really help in understanding this.

c) Here are my suggestions that the author might consider for the analysis:

Equation (1) makes sense, and it is understandable as an application of the chain rule, but it should be justified a little better as it is the basis of the whole analysis. The different parts of Fig. (1) seem to connect with Eq (1) but there is not a full correspondence.

How I see this, is that actually the analysis aims to look at the process chain from T->cloud prop, and also analyses some of the intermediate steps (T->OA, OA->CCN), but does not focus on the steps CCN->cloud prop. The forcing part ($dF/d(\text{cloud prop.})$) of Eq. 1 is completely ignored.

The authors would need to make a decision if they want to follow Eq(1) or not, and also structure the analysis along it. The deviations from equation (1) and fig(1) in the equations

should be explained, and possibly also the connection between the equation and the figure could be enhanced by adding the terms to the figure.

It is also possible to not follow the equation to the detail, but to say that Eq 1 is the basic equation that underpins the idea of the analysis, but that then different sub-parts are analyzed in detail from the data (e.g. $d\text{COT}/d\text{OA}$) to give insight on the process. It should, however, be made clear that the individual sensitivities are not analyzed in detail.

Once this decision is made, the description of the analysis could follow this plan, with each of the analyzed responses explained in the framework. It would be good if the reasoning behind Fig. (4) was explained more clearly, for example.

The authors can, of course, also choose an entirely different way of presenting the results. In any case, the presentation of Eq (1) as the basis of the analysis should either be updated or removed, as in current form this is not really followed.

Thanks for these great suggestions. We have followed them by editing both the illustration and the equations, thus making the analysis much easier to follow. We've also added explanatory text. The changes to the introduction are references above as well (L85-):

“The total feedback is by definition the change in radiative forcing (F) with temperature (T) and can be decomposed as follows:

$$\frac{dF}{dT} = \frac{d(\text{OA})}{dT} \cdot \frac{d(\text{CCN})}{d(\text{OA})} \cdot \frac{d(\text{cloud prop.})}{d(\text{CCN})} \cdot \frac{dF}{d(\text{cloud prop.})}$$

where CCN is cloud condensation nuclei concentration, OA is the organic aerosol mass, “cloud prop.” refers to cloud properties. In this study, we target the terms in the feedback up until changes in cloud properties, i.e.

$$\begin{aligned} \frac{d(\text{cloud prop.})}{dT} &= \frac{d(\text{OA})}{dT} \cdot \frac{d(\text{CCN})}{d(\text{OA})} \cdot \frac{d(\text{cloud prop.})}{d(\text{CCN})} \\ &= \frac{d(\text{OA})}{dT} \cdot \frac{d(\text{cloud prop.})}{d(\text{OA})} \end{aligned}$$

The relationship between changes in cloud properties and forcing ($\frac{dF}{d(\text{cloud prop.})}$) is currently an active area of research in its own right [e.g. 3, 42], which we, therefore, leave outside of the scope of this study. Note that using present-day conditions to evaluate the feedback limits our analysis to the “pure” temperature feedback and excludes the potential effect of CO₂ fertilization on BVOC emissions, which has been shown to be a large contributor in some ESMs [see e.g. 43]. We use number concentrations of particles larger than 50 nm, 100 nm and 200 nm (N_{50} , N_{100} , N_{200}) to investigate the link between OA and CCN, “cloud prop.” is investigated by changes in cloud droplet effective radius and cloud optical thickness, and we evaluate the terms $\frac{d(\text{CCN})}{d(\text{OA})}$ and $\frac{d(\text{cloud prop.})}{d(\text{OA})}$ both separately and combined, i.e. $\frac{d(\text{cloud prop.})}{d(\text{OA})}$, to assess the combined effect. In our process-based evaluation, we combine the insights from relevant observational data sets with the unrivalled ability of ESMs to produce projections on the global scale.”

The updated illustration is as follows:

We have further clarified in the text that we present $d(\text{cloud prop.})/d\text{OA}$ instead of $d\text{CCN}$ because $d\text{CCN}$ is hard to relate directly to the measurements/model output because it will depend on other properties. This is done through adding the following clarification at the start of section 3 in the results (L169, also referenced above):

“We here analyse the modelled and observed impact on cloud properties, as represented by cloud optical thickness and cloud droplet effective radius. By binning by cloud water path (CWP), we constrain the impact of different cloud regimes/types on our analysis and also effectively constrain it to mainly the cloud albedo effect (or the first indirect effect), leaving other aerosol-cloud interactions outside the scope of this study [3]. We present the change in cloud properties as a result of changes in OA, not CCN. This is because the activation diameter may vary both within and between models and reality, and to evaluate the feedback strength, it is easier to follow the signal as outlined in equation 3 by considering $d(\text{cloud props.})/d(\text{OA})$ rather than $d(\text{cloud props.})/d(\text{CCN})$. See the supplementary, section S6.2, for the same figure as 4 but with high versus low N100 and N50.”

** Soundness and detail of methodology:

The methodology is based on analysing two well-known, long-term datasets and a wealth of Earth System Model data. The data analysis methods are of sufficient quality, and choices made are generally well justified and also sufficient care has been taken to ensure that specific choices (such as the time period analyzed) are not unduly impacting results. The methodology is presented in sufficient detail and the datasets used are in principle such that the work can likely be reproduced.

Other questions and comments:

Title, abstract, and e.g. 1117: The authors use the term 'structural uncertainty in the model representation'. As one of the main claims is about structural uncertainty, maybe at least a

reference to how such and uncertainty is defined and evaluated (When is it large? How is it different from other uncertainties? How to address it) could be given for the reader.

This is a good point. Our main reason for claiming the error is structural is that the models have such anomalous relationships between temperature and organic aerosols in ATTO, indicating that the parameterizations may not be able to represent the processes. However, we realize that it is not possible to rule out that this is at least partly parametric error, based on the current analysis. We have therefore removed the “structural” from the title so it now reads: “Aerosol-cloud-climate feedbacks over forests: Clear evidence from observations, large uncertainty in Earth System Models” and further changed parts of the abstract to be:

“Based on observational evidence, there is large error in the representation of this feedback in the models, particularly for the tropical environment. The model evaluation shows that the weakest modelled feedback estimates can likely be excluded, but highlights compensating errors and possible structural uncertainties in the model representation.”

Abstract: On line 26, the author state that they use 'natural variability as a proxy for perturbed states of the climate'. This is not elaborated on in the paper at all. Either remove or explain this in more detail.

It is, in fact, mentioned in the introduction as well on current line 60 in the difference file: “However, the emergence of long-term in-situ observational data sets [20, 21, 22, 23, 24, 25, 26] gives rise to a unique opportunity to use natural variability in environmental parameters and aerosols as a proxy for perturbed states of the climate.” However, to make it clearer what is intended, we have also added an elaboration on (L81-):

“We evaluate the models by following the BSOA feedback from temperature via organic aerosol and size distribution and up until impacts on cloud properties [8] using natural variability in weather as a proxy for a perturbed climate state: i.e. we evaluate the relationships between the variables in the feedback loop under natural variability of weather”.

156: I think that currently "begging the question" is used as the authors use it, but some people might consider this bad English (see e.g. <https://www.theguardian.com/media/mind-your-language/2010/may/24/begging-the-question-mind-your-language>)

Fair, we have rephrased it to “This raises the question of what degree [...]”

1130: '...a similar pattern to the observed...' This seems very qualitative to me. Why is the pattern in Fig 4. significant, and how is the similarity evaluated? As this is the final and quite important step of the analysis, some more detail on the strength of the response was evaluated could be useful here. The same applies to line 136 ('...a reasonable response...'): what is reasonable? At which point would it become unreasonable?

This is a very valuable feedback and we have re-written the entire section to be more precise. Please see print screen and lines 183-215 in the revised manuscript.

189 The ESMS, on the other hand, do not provide a uniform picture of the response of the cloud properties to changes in OA.
 190 For SMEAR-II (Fig. 4, left), none of the analysed models consistently replicate the observed increase in COT and decrease
 191 in r_{eff} ~~with higher OA dayson days with high OA versus low OA~~. While NorESM produces the right sign for the difference
 192 in COT and r_{eff} , the magnitude of the response is clearly too high ~~(more than double that of the observations for CWP below~~
 193 ~~250 gm^{-2})~~ compared to the observations, ~~especially considering that the total OA concentrations (and thus also the change~~
 194 ~~between high and low) are lower in the model than in the observations~~. This is likely due to the overestimation of the slope
 195 between OA and N_{100} and N_{50} ~~meaning that the increase in CCN is too strong for high OA concentrations~~. The same analysis
 196 for N_{100} instead of OA (see supplementary figure S28a) also shows an equally strong overestimating, which could indicate
 197 a too strong aerosol sensitivity in general in the model. ECHAM-SALSA ~~has mostly a similar pattern to the observed, but~~
 198 ~~very weak response or even opposite sign for the lowest CWP bins~~ often is close to the observations in the median, but the
 199 uncertainties are high and the response is significantly different from zero for only very few CWP bins. The model also shows
 200 an increase in r_{eff} (as opposed to the expected decrease) for the smallest CWP bins at SMEAR-II, though not significantly
 201 different from zero. These results for ECHAM-SALSA stand in contrast to the same analysis for high versus low N_{100} and
 202 N_{50} (Fig. S28a, S28b, S29a and S29b), which show a stronger and more often significant response. It is therefore likely that the
 203 weak relationship between OA and N_{50} seen in the previous section (see Fig. 3) likely plays a significant role in reducing the
 204 impact of OA on cloud properties. UKESM and EC-Earth show a significant response only for the lowest CWP bins in the
 205 boreal environment, where the observations show ~~no a very weak~~ change in cloud properties. The low response in UKESM in
 206 the boreal zone is surprising given the strong relationships found in Figs. 2 and 3. ~~It turns out to not be~~ This is not due to a weak
 207 aerosol sensitivity though (see Fig. S28a), but is likely due to an erroneously low hygroscopicity of OA in UKESM (confirmed
 208 through code COTe inspection) which counteracts the effect of size during activation. ~~Note that the distribution of CWP in~~
 209 ~~UKESM is quite heavily skewed towards lower values, meaning that the two smallest bins showing a significant response in~~
 210 ~~4 for SMEAR-II actually constitute most of the data~~. EC-Earth has a similar lack of response in the higher CWP bins for
 211 N_{100} and N_{50} as for OA. This limits the impact of the feedback and is not in accordance with the observations, especially for the
 212 middle values of CWP. For ATTO, all the models underestimate the change in r_{eff} with elevated OA concentrations, possibly
 213 with the exception of ECHAM-SALSA. On the other hand, for COT, NorESM ~~shows a reasonable response, while is very~~
 214 ~~similar to the observations, though with some non-significant overestimation for the middle CWP bins~~. ECHAM-SALSA
 215 overestimates the response for all CWP bins and is more than a factor of two too high for all bins above 170 gm^{-2} . Note
 216 that both models show ~~somewhat~~ inconsistent responses in r_{eff} and COT compared to observations compared to COT, while
 217 the observations mostly show the response to be mirrored between the two. The low response of the modelled r_{eff} for all the
 218 models in ATTO might be related the droplet sizes being smaller in the models (see Fig. S32b), making the clouds less
 219 susceptible to perturbation (updraft- rather than CCN-limited) ~~-, [see e.g. 32])~~. This is consistent with the response of the cloud
 220 properties to high versus low N_{50} and N_{100} , which is similarly low as for OA, and also the fact that the models overestimate
 221 aerosol concentrations at ATTO. In NorESM in particular, ~~this would make sense due to~~ the extreme overestimation of both
 222 OA and number of particles in the region (Figs. 2 and 3), ~~leading to likely lead~~ the cloud regime to be updraft- rather than
 223 CCN-limited ~~(there are already enough CCN)-~~. Note that this is the opposite picture to SMEAR-II where most models were
 224 too sensitive to changes in N_{50} and N_{100} .

1200: is the word 'again' needed here?

No, good point, it has been removed.

Notes of other changes not directly in response to review comments:

For the plots in the model evaluation in the supplementary, EC-Earth was previously used in 3 hourly resolution, but this has now been updated to hourly resolution. It was previously in 3 hourly resolution because the temperature was used in the standard temperature and pressure (STP) conversion and the temperature is from IFS which only has 3 hourly resolution. To preserve the hourly output, we have now used quadratically interpolated temperature in the STP conversion. The impact on the results are negligible.

We have changed the model level to the second to bottom level in ATTO. This was already the case for UKESM, but we have now done it for all models because we wish to account for the fact that even though the bottom level would cover the measurement height under most conditions, the second level might be more representative due to boundary layer dynamics.

We corrected the timestamp in NorESM and UKESM with one hour due to time stamp issue (start of time period versus mid).

Note that the evaluation plots in the supplementary were by a mistake taken from the third height level of the model grid for all figures and this is why in particular ECHAM-SALSA has slightly higher concentrations in the newest version and the diurnal variations have changed somewhat.

The data for the analysis of temperature to OA to number concentration was filtered so that we use only days where we have more than 20/24 observations to avoid the influence of diurnal cycle.

References:

Abdul-Razzak, H. and Ghan, S.J. (2000) ‘A parameterization of aerosol activation: 2. Multiple aerosol types’, *Journal of Geophysical Research: Atmospheres*, 105(D5), pp. 6837–6844. Available at: <https://doi.org/10.1029/1999JD901161>.

Abdul-Razzak, H. and Ghan, S.J. (2002) ‘A parameterization of aerosol activation 3. Sectional representation’, *Journal of Geophysical Research: Atmospheres*, 107(D3), p. AAC 1-1-AAC 1-6. Available at: <https://doi.org/10.1029/2001JD000483>.

Fanourgakis, G.S. *et al.* (2019) ‘Evaluation of global simulations of aerosol particle and cloud condensation nuclei number, with implications for cloud droplet formation’, *Atmospheric Chemistry and Physics*, 19(13), pp. 8591–8617. Available at: <https://doi.org/10.5194/acp-19-8591-2019>.

Guenther, A. *et al.* (1995) ‘A global model of natural volatile organic compound emissions’, *Journal of Geophysical Research: Atmospheres*, 100(D5), pp. 8873–8892. Available at: <https://doi.org/10.1029/94JD02950>.

McCoy, I.L. *et al.* (2020) ‘The hemispheric contrast in cloud microphysical properties constrains aerosol forcing’, *Proceedings of the National Academy of Sciences*, 117(32), pp. 18998–19006. Available at: <https://doi.org/10.1073/pnas.1922502117>.

Pacifico, F. *et al.* (2011) ‘Evaluation of a photosynthesis-based biogenic isoprene emission scheme in JULES and simulation of isoprene emissions under present-day climate conditions’, *Atmospheric Chemistry and Physics*, 11(9), pp. 4371–4389. Available at: <https://doi.org/10.5194/acp-11-4371-2011>.

Pacifico, F. *et al.* (2012) ‘Sensitivity of biogenic isoprene emissions to past, present, and future environmental conditions and implications for atmospheric chemistry’, *Journal of Geophysical Research: Atmospheres*, 117(D22). Available at: <https://doi.org/10.1029/2012JD018276>.

Pacifico, F. *et al.* (2015) ‘Biomass burning related ozone damage on vegetation over the Amazon forest: a model sensitivity study’, *Atmospheric Chemistry and Physics*, 15(5), pp. 2791–2804. Available at: <https://doi.org/10.5194/acp-15-2791-2015>.

Yli-Juuti, T. *et al.* (2021) ‘Significance of the organic aerosol driven climate feedback in the boreal area’, *Nature Communications*, 12(1), p. 5637. Available at: <https://doi.org/10.1038/s41467-021-25850-7>.

REVIEWERS' COMMENTS

Reviewer #2 (Remarks to the Author):

Review of "Aerosol-cloud-climate feedbacks over forests: clear evidence from observations, large uncertainty in Earth System Models" by Blichner et al.

Thank you for providing me with the opportunity to review the revised version of this manuscript. I thank the authors for responding in detail to all my previous comments and am overall very pleased with the revisions made. The manuscript is now much clearer, the additional discussion points added are very welcome tying the findings together while making recommendations for future areas of work. My remaining comments below are all fairly minor/cosmetic and I do not need to rereview the paper once implemented. I am happy to recommend this revised version for publication in Nature Communications.

Minor comments (please note line number refer to line number in the Track Changed document):

L39: accurate/accurately used twice in the same sentence would remove one occurrence.

L83: I would rewrite this sentence as "We evaluate the models by examining the components of the BVOC-aerosol-climate feedback chain from the temperature dependence of emissions through to subsequent impacts on organic aerosol mass, size distribution and cloud properties."

L85/86 under natural variability of weather \diamond under natural environmental conditions [suggestion]

Figure S6/S11 plots OA vs isoprene emission relationship for all models, note that isoprene does not contribute to SOA in UKESM1, so any relationship here is by chance. The figures are therefore a bit misleading. I would either make this very clear in the supplementary text (for example you currently note the lower slope for UKESM on L75 Supplementary text) and/or remove the UKESM OA vs isoprene correlation from the relevant figures.

Figure 3 caption says currently NorESM values are divide by 4 but the title on the subplots for ATTO says NorESM/8?

L156 impacton  impact on

L163 by both due  both by an increased condensation sink and ...

L181 We focus of the  We focus on the

L200 overestimating  overestimation

L222 might be related the  might be related to

L234/235: Rather than referring to your “simplified picture” why not just refer to your “proposed feedback chain (Fig 1)”

L308 rule out it out  rule it out

Figure 5 add a legend to this figure illustration which colour bar corresponds to which model.

Table 1 , reference for UKCA-GLOMAP in UKESM model: note Mulcahy et al. (2020) is the most up-to-date reference for the aerosol representation in UKESM1 (<https://doi.org/10.5194/gmd-13-6383-2020>). Note this study in fact includes an evaluation of the seasonal cycle of OA mass concentration at SMEAR which is in pretty good agreement (see their Fig 7) with your assessment in Fig. S37. The authors in this paper highlight the important role of oxidant chemistry also by comparing a prescribed oxidant configuration with the full ESM with interactive oxidants and point to much higher oxidation in particular by NO₃ in the prescribed oxidant simulation. This would seem to be a relevant publication for this current study.

L435 MOSIS  MODIS

Reviewer #3 (Remarks to the Author):

The authors have addressed my concerns and the paper is now much better structure and now forms a convincing and logical presentation of the research and findings. As I already stated, the topic and content of the research are important for the scientific community and the paper is an original and noteworthy contribution to the field. Therefore, I propose accepting the paper for publication, but want to bring the following to the editor’s and author’s attention as at least the first point must be corrected (points 2 and 3 I leave at the Editor’s discretion).

1. In my version of the manuscript, Figures 2 and 3 are missing a significant portion of data (the majority of observation/model prediction points). The regressions are visible, and based on reading the paper, the data has not changed, but the editor should verify that the data are unchanged from the first version.

2. Line 103: “BVOC precursors” is a little ambiguous, as it could mean precursors of BVOC (in the biological synthesis pathways of BVOC emission, or (as I think is the meaning here) biogenic organic precursors of SOA.

3. Line 105: “while the β term is strictly related to the temperature dependency.” I think this is rather obvious (as β is mathematically describing the temperature dependency, and I think we are automatically strict about it), but I’m also wondering how to interpret this. I think that based on the given regressions that include heavy averaging, we cannot yet say that there would not be a temperature dependence in the yield or loss rates, which might also be temperature dependent but in a different way and not visible in this data. I propose adding rephrasing in a way that makes this clear.

Reviewer comments are in black, author responses are in blue.

Reviewer #2 (Remarks to the Author): _____ **1**
Reviewer #3 (Remarks to the Author): _____ **3**
Notes of other changes not directly in response to review comments: _____ **4**

Reviewer #2 (Remarks to the Author):

Review of “Aerosol-cloud-climate feedbacks over forests: clear evidence from observations, large uncertainty in Earth System Models” by Blichner et al.

Thank you for providing me with the opportunity to review the revised version of this manuscript. I thank the authors for responding in detail to all my previous comments and am overall very pleased with the revisions made. The manuscript is now much clearer, the additional discussion points added are very welcome tying the findings together while making recommendations for future areas of work. My remaining comments below are all fairly minor/cosmetic and I do not need to rereview the paper once implemented. I am happy to recommend this revised version for publication in Nature Communications.

We would like to again thank the reviewer for these comments, and we are very pleased that they are now happy with our edits.

Minor comments (please note line number refer to line number in the Track Changed document):

L39: accurate/accurately used twice in the same sentence would remove one occurrence.

Thanks, it now reads “To accurately capture these effects, reliable representation and evaluation of natural aerosol feedbacks in Earth System Models (ESMs) are needed.”

L83: I would rewrite this sentence as “We evaluate the models by examining the components of the BVOC-aerosol-climate feedback chain from the temperature dependence of emissions through to subsequent impacts on organic aerosol mass, size distribution and cloud properties.”

Thanks, great suggestion. It has been implemented.

L85/86 under natural variability of weather \diamond under natural environmental conditions [suggestion]

Thanks, this has been implemented!

Figure S6/S11 plots OA vs isoprene emission relationship for all models, note that isoprene does not contribute to SOA in UKESM1, so any relationship here is by chance. The figures are therefore a bit mis-leading. I would either make this very clear in the supplementary text (for example you currently note the lower slope for UKESM on L75 Supplementary text)

and/or remove the UKESM OA vs isoprene correlation from the relevant figures.
Figure 3 caption says currently NorESM values are divide by 4 but the title on the subplots for ATTO says NorESM/8?

This is a good point. We have added a sentence to the two captions in question: “Note that in UKESM, only monoterpene contributes to SOA formation.”.

We’ve added a sentence in the SMEAR-II discussion on L80: “Also note that in UKESM, isoprene does not contribute to SOA formation at all.”

We’ve also added made this clear in the discussion of the ATTO results with L112 now reading: “NorESM has a fairly strong relationship between emissions and OA at ATTO, while EC-Earth has a weak one and UKESM has a negative correlation for monoterpene (isoprene does not form SOA in UKESM).”

Thanks, the caption was wrong in Fig 4!

L156 impacton  impact on

Thank you, this has been fixed.

L163 by both due  both by an increased condensation sink and ...

Thank you, this has been fixed.

L181 We focus of the  We focus on the

Thank you, this has been fixed.

L200 overestimating  overestimation

Thank you, this has been fixed.

L222 might be related the  might be related to

Thank you, this has been fixed.

L234/235: Rather than referring to your “simplified picture” why not just refer to your “proposed feedback chain (Fig 1)”

Yes, good point!

L308 rule out it out  rule it out

Thank you, this has been fixed.

Figure 5 add a legend to this figure illustration which colour bar corresponds to which model.

Yes, good point!

Table 1 , reference for UKCA-GLOMAP in UKESM model: note Mulcahy et al. (2020) is the most up-to-date reference for the aerosol representation in UKESM1 (<https://doi.org/10.5194/gmd-13-6383-2020>). Note this study in fact includes an evaluation of

the seasonal cycle of OA mass concentration at SMEAR which is in pretty good agreement (see their Fig 7) with your assessment in Fig. S37. The authors in this paper highlight the important role of oxidant chemistry also by comparing a prescribed oxidant configuration with the full ESM with interactive oxidants and point to much higher oxidation in particular by NO₃ in the prescribed oxidant simulation. This would seem to be a relevant publication for this current study.

This is a good point! We have changed the reference in Table 1 accordingly.

L435 MOSIS  MODIS

This has been corrected accordingly.

Reviewer #3 (Remarks to the Author):

The authors have addressed my concerns and the paper is now much better structure and now forms a convincing and logical presentation of the research and findings. As I already stated, the topic and content of the research are important for the scientific community and the paper is an original and noteworthy contribution to the field. Therefore, I propose accepting the paper for publication, but want to bring the following to the editor's and author's attention as at least the first point must be corrected (points 2 and 3 I leave at the Editor's discretion).

1. In my version of the manuscript, Figures 2 and 3 are missing a significant portion of data (the majority of observation/model prediction points). The regressions are visible, and based on reading the paper, the data has not changed, but the editor should verify that the data are unchanged from the first version.

Thanks for noticing this. Firstly, only the ATTO data has changed.

There are two reasons for this (both noted at the end of the author replies):

- We have added a requirement for the number of datapoints per day to include a datapoint because we found during the extra analysis of emission rates that some datapoints were bias towards certain times of the day. We initially used 20 as a limit, but seeing the reviewers point that we do loose quite a few points, we have now changed this to 16, which is enough to avoid significant diurnal biases. We also realised we had not added this to the method, so we've also done this.
- The model data looks slightly different because we changed the level from the bottom level to the second to bottom level (does not affect the amount of data).

2. Line 103: "BVOC precursors" is a little ambiguous, as it could mean precursors of BVOC (in the biological synthesis pathways of BVOC emission, or (as I think is the meaning here) biogenic organic precursors of SOA.

We changed this to "BVOCs".

3. Line 105: "while the β term is strictly related to the temperature dependency." I think this is rather obvious (as β is mathematically describing the temperature dependency, and I think we are automatically strict about it), but I'm also wondering how to interpret this. I think that based on the given regressions that include heavy averaging, we cannot yet say that there would not be a temperature dependence in the yield or loss rates, which might also be

temperature dependent but in a different way and not visible in this data. I propose adding rephrasing in a way that makes this clear.

Thanks for this, we agree. The sentence now reads “[...] while the β term is related to the temperature dependency (including possible temperature dependencies of yields, oxidation, loss rates etc.).

Notes of other changes not directly in response to review comments:

Note that we discovered a mistake in the code during the final checks, and for the cloud properties plots (Fig 4 and parts of Fig 5), the UKESM data had not been properly selected for season. This issue has now been corrected, and does not significantly impact the results, though there is some change to Fig 4, especially in SMEAR-II. See below for comparison of new and old figures.

NEW VERSION Fig 4:

Difference between high OA and low OA

NEW VERSION Fig 5:

OLD VERSION Fig 5: